# Associative memory inspires improvements for in-context learning using a novel attention residual stream architecture

**Thomas F Burns**
*SciAI Center, Cornell University, USA*

**Tomoki Fukai**
*Neural Coding and Brain Computing Unit, OIST, Japan*

**Christopher J Earls**
*SciAI Center, Center for Applied Mathematics, School of Civil and Environmental Engineering, Cornell University, USA*

**Reviewed on OpenReview:** *https://openreview.net/forum?id=lcTFm4LIRR*

## Abstract

Large language models (LLMs) demonstrate an impressive ability to utilise information within the context of their input sequences to appropriately respond to data unseen by the LLM during its training procedure. This ability is known as in-context learning (ICL). Humans and non-human animals demonstrate similar abilities, however their neural architectures differ substantially from LLMs. Despite this, a critical component within LLMs, the attention mechanism, resembles modern associative memory models, widely used in and influenced by the computational neuroscience community to model biological memory systems. Using this connection, we introduce an associative memory model capable of performing ICL. We use this as inspiration for a novel residual stream architecture which allows information to directly flow between attention heads. We test this architecture during training within a two-layer Transformer and show its ICL abilities manifest more quickly than without this modification. We then apply our architecture in small language models with 8 million and 1 billion parameters, focusing on attention head values, with results also indicating improved performance at these larger and more naturalistic scales.

## 1 Introduction

Transformers (Vaswani et al., 2017) are a popular and performant class of artificial neural networks. Large Language Models (LLMs), decorated exemplars of Transformers, have demonstrated impressive capabilities in a wide array of natural language tasks (Brown et al., 2020). One particularly notable capability, known as in-context learning (ICL), has gained significant attention. ICL, as witnessed in LLMs, occurs when a model appropriately adapts to tasks or patterns in the inference-time input data which was not provided during the model's training procedure (Lynch & Sermanet, 2021; Mirchandani et al., 2023; Duan et al., 2023). This ability to immediately learn and generalise from new information, especially with only a single or few exposures to new information, is a hallmark of sophisticated cognitive abilities seen in biological systems; human and non-human animals are capable of rapidly adapting their behaviour in changing contexts to achieve novel goals (Miller & Cohen, 2001; Ranganath & Knight, 2002; Boorman et al., 2021; Rosenberg et al., 2021), and can infer and apply previously-unseen, and even arbitrary rules without significant learning periods (Goel & Dolan, 2000; Rougier et al., 2005; Mansouri et al., 2020; Levi et al., 2024). Developing models rooted in computational neuroscience, such as associative memory, to connect our understanding of related phenomena in both artificial and natural intelligence may provide valuable insights for developing more adaptive and versatile language models, in addition to helping us more deeply understand more general

neural systems (see Appendix A for discussion on the connection between associative memory models and modern neuroscience).

While various explanations have been proposed for how LLMs learn to perform ICL (Olsson et al., 2022; Von Oswald et al., 2023; Li et al., 2023; Reddy, 2024), there has been little work to develop explanations which also offer neurobiologically-plausible models of similar abilities seen in humans and non-human animals. One exception to this is the recent work of Ji-An et al. (2024), which illustrates a connection between ICL in LLMs and the contextual maintenance and retrieval model of human episodic memory from psychology. This model proposes that memory items are stored as a contextual composition of stimulus- and source-related content available during, and near-in-time, to a memory item's presentation. The model comports with several reported psychological phenomena in humans (Lohnas et al., 2015; Cohen & Kahana, 2022; Zhou et al., 2023), and can be constructed using a Hebbian learning rule (Howard et al., 2005). Hebbian learning (Hebb, 1949), that neurons which 'fire together, wire together', is foundational to the theoretical underpinnings of associative memory models (Nakano, 1972; Amari, 1972; Little, 1974; Hopfield, 1982), which propose that memory items are stored by strengthening the connections between neurons which become activated during and near-in-time to the stimulus corresponding to memory items' presentations. How associative memory is neurophysiologically implemented is well-studied (Amit, 1990; Buzsáki, 2010; Khona & Fiete, 2022; Burns et al., 2022), and this is complemented by a well-developed theoretical literature, including work noting the close resemblance to a core ingredient of LLMs, the attention mechanisms of Transformers (Ramsauer et al., 2021; Bricken & Pehlevan, 2021; Kozachkov et al., 2022; Burns & Fukai, 2023; Burns, 2024; Gershman et al., 2025).

## 1.1 Contributions

Given these existing links, further developing connections between the framework of associative memory and ICL may offer deeper insights or improvements. In the following sections, we:

- introduce a one-layer associative memory model which can perform ICL on a classification task, and which analogously allows attention values to directly represent input data;

- using the same task, and inspired by our explicit associative memory model, show how creating a residual stream of attention values between attention heads in a two-layer Transformer speeds-up ICL during training (compared to the vanilla Transformer and applying the same technique to queries or keys); and

- demonstrate that naïvely applying the same idea in small language models (LMs) indicates performance improvements scale to larger models and more naturalistic data.

A central theme in our innovation is a focus on the role of values in the attention mechanism, and architecting a simple 'look-back' method in the form a residual connection of values. Residual connections, also known as 'skip' or 'shortcut' connections, can be described as those which connect neurons which are otherwise indirectly connected through a more prominent pathway. First identified in experimental neuroscience (Lorente de Nó, 1938) and considered since the dawn of theoretical neuroscience and artificial neural networks (McCulloch & Pitts, 1943; Rosenblatt, 1961), researchers continue to find residual connections useful in modern applications (Dalmaz et al., 2022; Huang et al., 2023; Zhang et al., 2024b). A noticeable feature of Transformers is its use of the so-called *residual stream*, wherein data, once processed by the attention and feedforward layers, is added back to itself. What can therefore be considered a 'cognitive workspace' (Juliani et al., 2022) has been shown to contain rich structure, amendable to popular (Elhage et al., 2021) and emerging (Shai et al., 2024) interpretability methods. Our work illustrates that specific additional residual connections can lead to enhanced performance in ICL tasks, and we speculate it may also aid in future interpretability efforts.

## 2 ICL classification with a one-layer associative memory network

### 2.1 ICL classification task and mathematical set-up

Let $X \in \mathbb{R}^{e \times \jmath}$ be the input sequence data, where $e$ is the dimension of each token embedded within a suitable latent space, and $\jmath$ is the number of tokens in the sequence[1]. Each token is considered either an *object*, $o$, or *label*, $l$. Input $X$ consists of a sequence of multiple pairs of objects and labels. We say a *pair* of tokens is a contiguous sub-sequence of two column vectors from within $X$, consisting of one object token followed by one label token. For example, the $j$-th pair $X' \in \mathbb{R}^{e \times 2}$ consists of one object token $o^j \in \mathbb{R}^e$ followed by one label token $l^j \in \mathbb{R}^e$. Our input sequence $X$ will therefore consist of multiple pairs, and each pair may appear more than once. The final token of a sequence, denoted as $x_\jmath$, corresponds to the label token of the last pair, which itself has appeared at least once prior to this final instance in $X$. However, the true label token data of this final label token is replaced with the zero vector, and the network's task is to correctly predict this true label token given the previous in-context instance of the tokens' data (as illustrated in Figure 1a).

The token embeddings, representing objects and labels, follows Reddy (2024), where all token embeddings are drawn from similar statistical distributions. For every instance of a pair, label token embeddings come from fixed vectors, whereas object token embeddings are constructed from combinations of fixed and random vectors. Each label token $l^i$ is an $e$–dimensional vector $\mu_{l^i}$, whose components are i.i.d. sampled from a normal distribution having mean zero and variance $1/e$. Each object token embedding, $o^i$, is given by

$$o^i := \frac{\mu_{o^i} + \varepsilon \eta}{\sqrt{1 + \varepsilon^2}},$$

where $\mu_{o^i}$, which is fixed across all instances in $X$ of the pair, and $\eta$, which is drawn randomly for each instance in $X$ of the pair, are $e$–dimensional vectors whose components, like $\mu_{l^i}$, are i.i.d. sampled from a normal distribution having mean zero and variance $1/e$. The variable $\varepsilon$ controls the inter-instance variability of objects and, in the following, is set to 0.1 unless otherwise stated. This means that the final, tested instance of a pair has, as its object token, a slightly different appearance than the previous instance(s) seen in $X$ and is not a perfect match, where $\mu_{o^i}$ provides the commonality between these variations. Adding these variations makes the task less trivial and slightly more naturalistic.

In §2.2, we show it is possible to perform ICL with such pairs in a single forward step of a one-layer associative memory network, written in the language of a single Transformer attention head (Vaswani et al., 2017). To show this, and for the benefit of subsequent sections, we briefly summarize the classical Transformer set-up. In Transformers, each attention head consists of learnt parameters – weight matrices $W^q, W^k \in \mathbb{R}^{\hbar \times e}$ and $W^v \in \mathbb{R}^{\nu \times e}$ – with which, when taken together with the input data sequence $X$, we calculate the queries $Q$, keys $K$, and values $V$ matrices using

$$Q = W^q X, \quad K = W^k X, \quad \text{and} \quad V = W^v X.$$

The values $\hbar$ and $\nu$ are the reduced embedding dimensions for the attention operation (*i.e.*, to facilitate multi-headed attention, *etc.*). Here we use $\hbar = \nu$. As in the input data, $e$ is the dimension of the unreduced embedding space of the tokens.

It is then useful to define the SOFTMAX function for matrix arguments. For a matrix $M \in \mathbb{R}^{c \times \iota}$, we write $t_i := M[i,:] \in \mathbb{R}^{\iota}$ for the $i$-th row and $t_j := M[:,j] \in \mathbb{R}^{c}$ for the $j$-th column. Then, we define the SOFTMAX function for a matrix $M$ as $\text{SOFTMAX}(M)[t_i, t_j] := \frac{\exp(M[t_i, t_j])}{\sum_t \exp(M[t, t_j])}$, where $i$ and $j$ are the vector component indices. We use this to compute attention-based embeddings of the data $X$, denoted by $\tilde{X}$, as

$$\tilde{X} = \text{SOFTMAX}\left(\frac{1}{\sqrt{\hbar}} K^T Q\right) V, \tag{1}$$

in which we refer to the term $K^T Q$ as the *scores* $S \in \mathbb{R}^{\jmath \times \jmath}$. In Transformers and Transformer-based models such as LLMs, this new data $\tilde{X}$ is then recombined with data from other attention heads, before passing through a multi-layer perceptron. Layers of attention heads and multi-layer perceptrons are stacked atop one-another to perform increasingly sophisticated computations.

---

[1]We provide notation tables in Appendix E.

## 2.2 Associative Memory for ICL (AMICL) model

In our associative memory model, which we call *Associative Memory for ICL* (AMICL), we take inspiration from the transformations applied to the input data $X$ to create the basis of what can be considered (Ramsauer et al., 2021) as an associative memory update step in Equation 1. Instead of using parameterised values, we use a simple set of assignments for each token's key, query, and value vectors. For all embedded tokens $i < \mathfrak{s}$, we set $k_i = q_i = \frac{ax_{i-1} + x_i}{a+1}$ as the keys and queries, where lowercase Latin letters denote indexed column vectors, taken from the matrices that are denoted with uppercase Latin letters. For simplicity, we wrap the indices along the token sequence such that the first token, $x_1$, and the last token, $x_{\mathfrak{s}}$, are used to generate the sub-sequence $(x_{\mathfrak{s}}, x_1)$, *i.e.*, we assign token index 0 to $\mathfrak{s}$. For token $\mathfrak{s}$, with $x_{\mathfrak{s}}$ as the token column vector, we set $q_{\mathfrak{s}} = \frac{ax_{\mathfrak{s}-1} + x_{\mathfrak{s}}}{a+1}$ as the query and $k_{\mathfrak{s}}$ as the zero vector, which acts as its key. For all tokens, the values column vector is equal to the token column vector, *i.e.*, $v_j = x_j$. The value of $a$ can be any arbitrary real value, but, after testing within the range $[0, 2]$, is set as $a = 2$. (The justifications for testing $a$ within this range and setting $a = 2$ is given later in this subsection.)

We then perform next token prediction using the universal associative memory framework (Millidge et al., 2022), which can be interpreted as a generalisation of Equation 1. In particular, Equation 1 in the universal associative memory framework has the form PROJECTION(SEPARATION(SIMILARITY($K, Q$)), $V$), where the SIMILARITY function is chosen as a scaled dot product, the SEPARATION function is chosen as a SOFTMAX, and the PROJECTION function is chosen as a product.

By constructing the queries and keys in the AMICL model using what is essentially an 'average' between the current token and the previous token, with a scalar weighting of $a$ on the previous token, we allow the similarity function to identify the relevant pair in the context (in a similar sense to that of a 1D convolution operation), which is then potentially amplified by the separation function (*e.g.*, in the SOFTMAX case), for projection to the appropriate matching final token. In this way, and as illustrated in Figure 1, the AMICL model essentially performs auto-association on the final pair of tokens, treating the penultimate token as a context clue for the final token (in the same way that auto-associative memory dynamics traditionally use partial memory identity information to complete the full, remaining pattern information (Nakano, 1972; Amari, 1972; Little, 1974; Hopfield, 1982)).

Normatively, this design reflects the goal of the ICL task: to recall or reconstruct the most appropriate output token given its prior contextual association. Using the identity function ensures that what is retrieved is the token itself, not a transformation of it, mirroring the associative memory setting where retrieval should produce the original pattern.

The reason for choosing the values to have the identity of the original data, then, is because we cast this ICL task as an auto-association problem, and by the universal associative memory framework (Millidge et al., 2022), the projection function in the case of auto-association is the identity function (also see §3.1 of Millidge et al. (2022)).

From a biological perspective, this is also consistent with Hebbian learning in attractor networks, where activation patterns corresponding to past stimuli are reactivated as-is during recall (Nakano, 1972; Amari, 1972; Little, 1974; Hopfield, 1982), rather than being replaced by arbitrary transforms.

This then justifies the choice of exploring the range of $a$ in the positive reals only, since although negative values are mathematically possible, they would create an anti-Hebbian or repelling force, counter-acting the designed (Hebbian) auto-association task created by the query and key constructions.

As an intuitive sketch, AMICL can be thought of as implementing the algorithm illustrated in Figure 1: first, we identify the final pair, where the final token, $x_{\mathfrak{s}}$, which should be a label token, has been set to the zero vector (Figure 1a, where '?' represents the unknown token data and other tokens' data are shown); second, we consider all possible contiguous pairs in the context, *e.g.*, $(x_3, x_4), (x_4, x_5), \ldots$, without knowledge of their data, which we as designers know will correspond to pairs like, *e.g.*, $(o^2, l^2), (l^2, o^3), (o^3, l^3), \ldots$ (Figure 1b, where each contiguous pair is enclosed by a yellow rectangle); third, we compare all of the contiguous pairs in the second step with the final pair identified in the first step (which we are attempting to 'pattern complete'), and, upon seeing which context pair is most similar in final pair (in this example, $(o^1, l^1)$), complete the pattern appropriately by setting $x_{\mathfrak{s}} = l^1$ (Figure 1c).

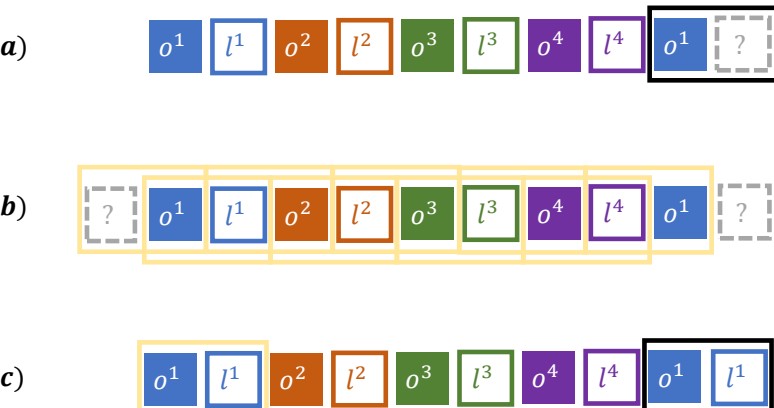

Figure 1: Depiction of the AMICL algorithm on label–object pairs, consisting of three steps: (a) consider the penultimate and final tokens as a 'local pattern' which has been 'partially corrupted' by the missing data in the final token, illustrated by the black rectangle enclosing this final pair; (b) search for matching, 'complete local patterns' by grouping all previous contiguous token pairs in the context, with each pair illustrated by a yellow rectangle enclosing the tokens; and (c) complete the final, corrupted local pattern based on matching to the nearest complete in-context pair, which in this case is $(o^1, l^1)$, so the final token data is assigned $l^1$.

Varying the parameter $a$ between 0 and 2 for combinations of similarity and seperation functions showed that using the DOT PRODUCT similarity with ARGMAX separation function resulted in practically perfect ICL ability at $a \geq 1.5$ whereas using PEARSON'S CORRELATION similarity with ARGMAX separation function saturated at a performance level of $\sim 85\%$ accuracy for the ICL pairs task (see Figure 12 in Appendix C). Due to the practically perfect ICL performance in the DOT PRODUCT case and the saturation of ICL performance PEARSON'S CORRELATION – both saturating at values at $a \geq 1.5$ – we set $a = 2$.

While setting $a = 2$ and PROJECTION = IDENTITY, we tested all combinations of:

- SIMILARITY $\in \{$DOT PRODUCT, PEARSON'S CORRELATION, MANHATTAN DISTANCE, EUCLIDEAN DISTANCE$\}$; with

- SEPARATION $\in \{$IDENTITY, SOFTMAX, ARGMAX$\}$.

Manual inspection of the resulting attention matrices indicated that the DOT PRODUCT and PEARSON'S CORRELATION similarity functions combined with the ARGMAX separation function provided the cleanest ICL (see Figure 11 in Appendix C for an example). Varying the token sequence length $s$ between 10 and 1,000 shows no variation in performance; varying the embedding dimension of the tokens $e$ between 10 and 1,000 showed that for $e \geq 50$, the ICL task performance was practically perfect (see Figure 13 in Appendix C).

## 3 Residual attention streams in a two-layer Transformer

### 3.1 Residual attention streams

In the AMICL model, following the path of information from the input to the queries, keys, and values (as illustrated in Figure 2a), we can see that the queries $Q$ and keys $K$ flow from a shared function $f$ of the input $X$. Whereas, the values $V$ are given directly by the input $X$. In the language of the universal associative memory framework (Millidge et al., 2022), we can say that the *similarity* and *separation* factors (determined by the queries $Q$ and keys $K$) are coupled by the shared function $f$ whereas the *projection* factor (given by the values $V$) is simply the input itself, *i.e.*, auto-association.

Seen through the lens of the traditional self-attention mechanism, this implies that a similar construction of the queries and keys is possible, such that using a simple identity function from the input for the values could replicate this behaviour in more sophisticated data, task, and model scenarios. However, in our initial experimentation with this idea, whereby we eliminated the $W^v$ matrices altogether, we witnessed significant model performance degradation and training instability (see Appendix B) – it also principally limits the model's expressiveness in the self-attention mechanism to auto-association, whereas many interesting tasks make use of some amount of *hetero*-association – see Burns (2024). This led us to consider an alternative form of projection, one which could in principle mix auto- and hetero-association, via the creation of values $V$ which more explicitly retain the prior input $X$. In particular, we introduce a residual connection between the values data of successive layers in the Transformer, which we call a *residual values stream* and, more generally, a *residual attention stream* applied to values (Figure 2b).

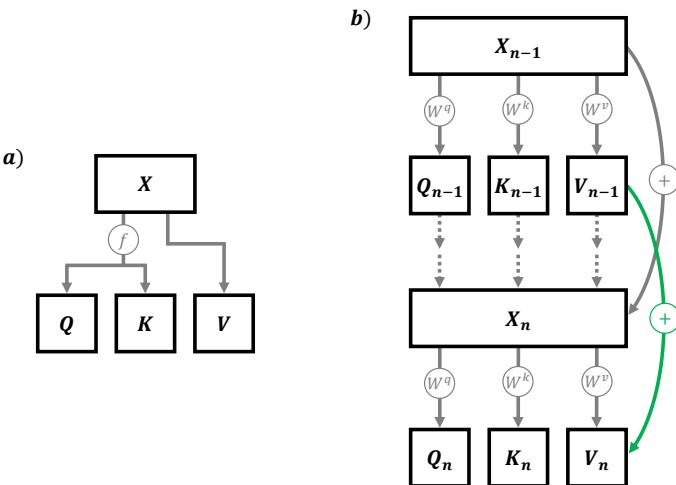

Figure 2: Partial diagrams of (a) the AMICL model and (b) our residual attention stream architecture with two Transformer layers, $n$ and $n-1$, shown here implemented for values (the added residual is shown in green). Boxes represent variables. Full arrows represent functions, with overlaid circles indicating relevant variables or functions (no circle is used for the identity function). Dotted arrows represent functions and variables omitted in the diagram for space.

As an alternative source of intuition, one can consider the informal interpretation of self-attention as consisting of: queries 'asking each token a question about other tokens'; keys 'responding to each token with the answer to the queries' questions', *i.e.*, they align with or 'attend to' the queries; and values giving the (weighted) answers to those questions, which are operationalised as small additions to the current token vectors. Within this informal conceptual model, we can consider our residual attention streams as retaining additional 'look-back' information in the answers.

Our residual attention stream also generates additional gradient signals during training, which could themselves be beneficial for the network to develop ICL capabilities. We therefore also test the same residual stream architecture, separately, for queries and keys. In principle, it is also possible to apply the residual stream architecture to combinations of queries, keys, and values. But, to maintain a stronger connection to the AMICL model, we focus on testing the residual value stream separately, as well as the keys and queries, again separately, for comparison.

### 3.2 Two-layer Transformer architecture

We train classic and modified versions of a two-layer Transformer on the same task described in Section 2. Following Reddy (2024), the common architectural features between the two versions consist of two single-head attention layers followed by a three-layer multi-layer perceptron with 128 ReLU neurons followed by a softmax layer to give probabilities over the $\ell$ labels. The network is trained with the same task as the AMICL model, using the cross-entropy loss between the predicted and actual final label in token $x_\jmath$. Within the modified architecture, we add the first attention head's queries, keys, or values to the second attention head's queries, keys, or values, respectively. More formally, in our modified version of the Transformer architecture, we calculate the first attention layer as

$$Q_1 = W_1^q X, \quad K_1 = W_1^k X, \quad \text{and} \quad V_1 = W_1^v X,$$

and then, in the second attention layer, for a residual queries stream we calculate

$$Q_2 = W_2^q X + Q_1, \quad K_2 = W_2^k X, \quad \text{and} \quad V_2 = W_2^v X,$$

or for a residual keys stream we calculate

$$Q_2 = W_2^q X, \quad K_2 = W_2^k X + K_1, \quad \text{and} \quad V_2 = W_2^v X,$$

or for a residual values stream (shown in Figure 2b) we calculate

$$Q_2 = W_2^q X, \quad K_2 = W_2^k X, \quad \text{and} \quad V_2 = W_2^v X + V_1.$$

### 3.3 Supplemental ICL tasks

As in Reddy (2024), while we train on the originally-described ICL task, we create supplemental tasks which are not used for training but rather act as proxy measurements of progress for different computational strategies for completing the task: training data memorisation and ICL capability generalisation. Namely, these supplemental tasks are:

- In-weights (IW): a series of object–label pairs is presented where the final pair is not found within the prior context but is present in the training data;

- In-context (IC): a series of novel object–label pairs, not seen in the training data (*i.e.*, an entirely re-drawn set of $\mu_{o^i}$ and $\mu_{l^i}$ values) but following exactly the same statistical structure, is presented; and

- In-context 2 (IC2): a series of object–label pairs are presented, where the objects are found in the training data but have been assigned new labels (*i.e.*, the objects retain their $\mu_{o^i}$ values but the labels have their $\mu_{l^i}$ values re-drawn).

The reduced embedding dimensions of the attention operation are both set to 128, *i.e.*, $\hbar, v = 128$. Each network architecture was trained on four random seeds, with a batch size of 128, vanilla stochastic gradient descent, and a learning rate of 0.01.

### 3.4 Performance of the attention residual streams on the ICL classification task

Figure 3 shows the accuracies for the task being trained (Test) and the three supplemental tasks (IW, IC, and IC2) for each architecture. It also shows the Test loss for all architectures and all loses for the values residual stream architecture. For our residual attention stream modifications, we observe a general leftward shift in all but the IW task, with the value residual stream architecture showing the largest shift. The IW task also shows a slower learning rate, which can be attributed to the relative difficulty of the network memorising the training data.

To quantify these shifts, we report statistics in Table 1 of when the first training snapshots we recorded reached an accuracy threshold of $> 0.95$. We find the values stream networks perform best, reaching the

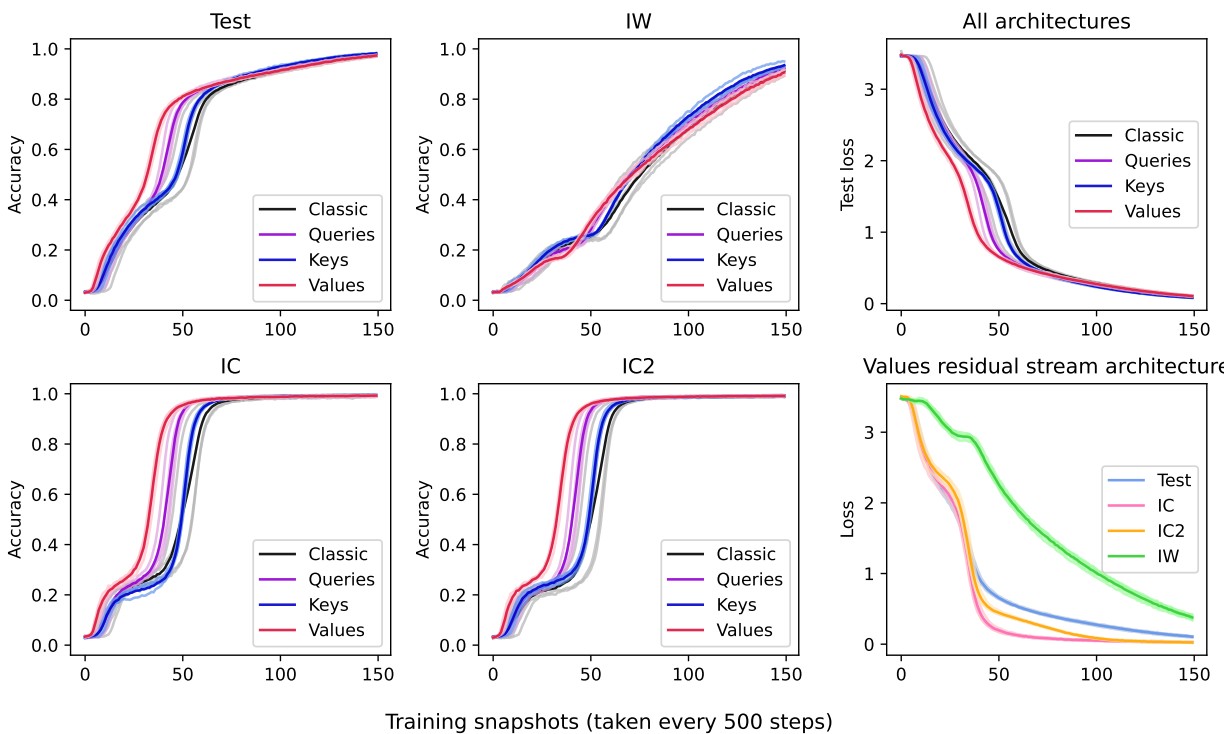

Figure 3: Accuracies and losses for the Test, IW, IC, and IC2 tasks over training time for the classic (unmodified) network and residual queries, keys, and values stream networks. Dark lines represent means, light lines represent individual trials.

same level of accuracy as the classic (unmodified) networks with $\sim 24\%$ fewer training steps. The same result is also seen at lower thresholds (see Tables 5 and 6 in Appendix D for thresholds of 0.5 and 0.9, respectively). We also performed t-tests at the 0.95 accuracy threshold, which showed significant differences between the performance of the classic (unmodified) network and the residual queries and values stream networks on the IC and IC2 tasks ($p < 0.01$, $t < -3.9$), but not between the classic and residual keys stream networks ($p > 0.39$, $t > -1.0$). The residual values stream networks also perform significantly better ($p < 0.03$, $t < -3.0$) than all other networks on both the IC and IC2 task, except the residual queries stream network for the IC task ($p = 0.06$, $t = -2.33$).

Table 1: Mean $\pm$ standard deviation of training snapshot number where accuracy first exceeded 0.95 for the IC and IC2 tasks in the classic (unmodified), and residual queries, keys, and values stream networks. Fastest training times are bolded.

|     | Classic | Queries (ours) | Keys (ours) | Values (ours) |
|-----|---------|----------------|-------------|---------------|
| IC  | $64.5 \pm 5.12$ | $52.5 \pm 1.5$ | $61.75 \pm 0.83$ | $\mathbf{49.0} \pm 2.12$ |
| IC2 | $63.25 \pm 4.15$ | $51.75 \pm 1.92$ | $61.25 \pm 0.83$ | $\mathbf{47.75} \pm 1.3$ |

## 4 Residual value streams in a toy language model

### 4.1 Toy language model architecture and training details

To provide an initial indication as to whether the benefit for ICL performance seen in Section 3 scales to larger models and naturalistic data, we also test the residual value stream architecture by training toy LMs. The Transformer-based model contains approximately 8 million parameters spread over eight layers, each with 16 attention heads. The context window is 256 tokens long and dimension of the model is 256, *i.e.*,

each token is mapped to a 256-dimensional vector. Next-token prediction is trained for three epochs on the Tiny Stories dataset, such that they can generate simple and short children's stories (Eldan & Li, 2023). Each network architecture was trained in three separate instances, using different random seeds for each instance but controlling for randomness between architectures by re-using the same set of seeds for the two architectures.

We implement the residual value stream in a naïve way: in each but the first layer, the values of each attention head receives an additional input of the values from the attention head with the same index in the previous layer[2]. Although there are alternative implementations of such residual streams, we chose a naïve and straightforward approach to provide an initial indication of the potential scalability of this architectural change and to not alter the number of learnt parameters between the networks.

The residual values stream networks achieved $1.65 \pm 0.05$ training loss and $1.55 \pm 0.01$ validation loss, slightly lower than the training and validation losses for the classic networks, which were $1.67 \pm 0.05$ and $1.56 \pm 0.01$, respectively. Measured by wall clock computation time[3], the residual values stream networks also took slightly longer to train, $30.18 \pm 0.49$ hours compared to $29.12 \pm 0.40$ hours for the classic networks. This additional computation time is attributable to the additional gradient computations required by the 16 additional residual streams connecting the attention head values at each but the first layer.

## 4.2 Evaluation of ICL ability using a simplistic natural language task

As a proxy evaluation of the ICL ability of each model in natural language, we utilise a simple indirect object identification (IOI) task. In natural language, a direct object is the noun that receives the action of the verb and an indirect object is a noun which receives the direct object, *e.g.*, in the sentence "A passed B the ball": "passed" is the verb; "A" is the subject; "the ball" is the direct object; and "B" is the indirect object. In the IOI task, we test for correct completion of sentences like "When A and B were playing with a ball, A passed the ball to", where " B" is the indirect object and is considered the correct completion.

GPT-2 Small can perform instances of the IOI task, and the circuit responsible for this ability has been identified (Wang et al., 2023). As shown in this section, our toy LMs have some measurable ability to complete instances of this task, and so we use this to compare the performance of the classic (unmodified) and residual value stream architectures.

For the IOI task, we tested the following sentences:

1. "When John and Mary went to the shops, $*$ gave the bag to", where $*$ was either "John" or "Mary", with correct completions " Mary" and " John", respectively.

2. "When Tom and James went to the park, $*$ gave the ball to", where $*$ was either "Tom" or "James", with correct completions " James" and " Tom", respectively.

3. "When Dan and Emily went to the shops, $*$ gave an apple to", where $*$ was either "Dan" or "Emily", with correct completions " Emily" and " Dan", respectively.

4. "After Sam and Amy went to the park, $*$ gave a drink to", where $*$ was either "Sam" or "Amy", with correct completions " Amy" and " Sam", respectively.

For each sentence and variation thereof (swapping the identities of the subject and indirect object, as indicated by the $*$ symbol above), we recorded the next token probabilities of the correct and incorrect names. As summarised in Table 2, we find the classic networks are much less capable than the residual values stream networks – across the four sentences, the classic networks correctly identify the indirect object with a probability of $\sim 7\%$ while the residual values stream networks do so with a probability of $\sim 41\%$, a $\sim 590\%$

---

[2]Principally, the choice of which attention heads form the residual values stream is arbitrary given the independent nature of each head. However, when implemented with multi-head attention, it is preferable to use the same head index for computational convenience.

[3]Using a PC equipped with an AMD Ryzen Threadripper PRO 3975WX 32-Cores CPU, 130GB of RAM, and $4\times$ NVIDIA RTX A4000 GPUs.

improvement. Similarly, the classic networks more regularly mistake the subject for the indirect object with a probability of $\sim 5\%$ while the residual values stream networks do so with a probability of $\sim 3\%$, a $\sim 60\%$ reduction.

Table 2: Mean $\pm$ standard deviation probabilities (%) of correct and incorrect responses to each sentence for the IOI task for the classic (unmodified) and residual values stream networks. Best scores are bolded.

| *Classic* | Sentence 1 | Sentence 2 | Sentence 3 | Sentence 4 |
|---|---|---|---|---|
| Correct (higher is better) | $11.73 \pm 15.62$ | $11.88 \pm 9.09$ | $3.83 \pm 6.09$ | $1.54 \pm 2.31$ |
| Incorrect (lower is better) | $6.93 \pm 5.70$ | $7.99 \pm 9.52$ | $0.51 \pm 0.66$ | $4.86 \pm 8.35$ |
| ***Residual values stream (ours)*** | Sentence 1 | Sentence 2 | Sentence 3 | Sentence 4 |
| Correct (higher is better) | $\mathbf{42.89} \pm 10.58$ | $\mathbf{43.44} \pm 14.45$ | $\mathbf{49.24} \pm 9.89$ | $\mathbf{28.56} \pm 5.92$ |
| Incorrect (lower is better) | $\mathbf{5.68} \pm 3.76$ | $\mathbf{6.8} \pm 10.40$ | $\mathbf{0.03} \pm 0.03$ | $\mathbf{0.18} \pm 0.15$ |

We propose the following explanations as for why the residual values stream variant outperforms the classic model so handedly despite the two models having very similar final loss values and the former having only a slightly smaller loss:

- especially in small models, even small differences in the training loss can translate to large differences in the test accuracy on downstream tasks (Lau et al., 2023);

- algorithmically, models with only very small or even no loss differences can perform the task in principally different ways (Power et al., 2022; Bushnaq et al., 2024); and

- provision of the residual values stream improves sample complexity for the IOI task, which while implicitly present in the auto-regressive training set-up, is not what is primarily measured by the training loss (which is more generally measuring performance on next-token prediction, implicitly consisting of a much larger variety of natural language tasks).

# 5 Residual value streams in a small language model

## 5.1 Small language model architecture and training details

To test the performance of residual value streams on a larger scale, we trained two one-billion parameter (1B) language models in the style of a Llama 3 model (Grattafiori et al., 2024). These models were trained on English-language documents, amounting to approximately 84 billion tokens, from a random subset of the Nemotron-CC-HQ dataset (Su et al., 2024a). As in the toy language model described in §4, the only way in which these models differed was in the presence or absence of the residual values stream. As in the 8M-parameter models, the 1B models showed no significant difference in training loss (see Figure 4).

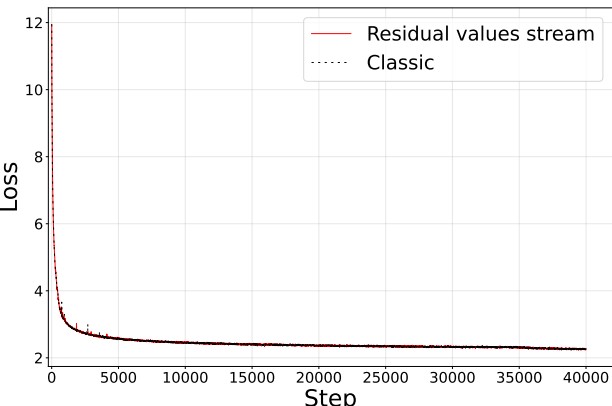

Figure 4: Training loss over steps for the 1B models.

Each model comprised of 16 layers, 2,048 hidden dimensions, 32 attention heads, and eight key-value heads. These models support a context length of 4,096 tokens and incorporate several modern architectural features: rotary positional embeddings (RoPE) (Su et al., 2024b) with a base of 500,000, SwiGLU activation functions (Shazeer, 2020) with a multi-layer perceptron (MLP) expansion factor of four, and RMSNorm for layer normalisation (Zhang & Sennrich, 2019).

The model was trained for 40,000 iterations with a global batch size of 512. We used the AdamW optimizer (Loshchilov & Hutter, 2019) with $\beta_1 = 0.9$, $\beta_2 = 0.95$, $\epsilon = 1e-8$, and gradient clipping at 1.0. The learning rate followed a cosine decay schedule with an initial rate of 3e-4, 500 warmup steps, 5,000 cooldown steps, and minimum learning rate of 0.0. The model was trained using the Llama-3.1-8B tokenizer (Grattafiori et al., 2024), which has a vocabulary size of 131,072 tokens. We used BFloat16 precision throughout.

Tables 3 and 4 summarise the training parameters.

| $N_\text{vocab}$ | $n_\text{Layers}$ | $n_\text{heads}$ | $d_\text{model}$ | $d_\text{MLP}$ | Batch Size | Sequence Length |
|---|---|---|---|---|---|---|
| 131,072 | 16 | 32 | 2,048 | 8,192 | 512 | 4,096 |

Table 3: Hyper-parameters used in the 1B models.

| Optim. | $\beta_1$ | $\beta_2$ | $\epsilon$ | $N_\text{Warmup}$ | $N_\text{Cooldown}$ | learning rate |
|---|---|---|---|---|---|---|
| AdamW | 0.9 | 0.95 | 1e-8 | 1% | 10% | 3e-4 |

Table 4: Optimizer parameters for the 1B models.

## 5.2 Small language model evaluation and results

After training these models, we evaluated them on single- and five-shot language understanding tasks, namely: AI2 Reasoning Challenge (ARC) (Clark et al., 2018), Physical Interaction: Question Answering (PIQA) (Bisk et al., 2019), OpenBookQA (Mihaylov et al., 2018), and HellaSwag (Zellers et al., 2019).

Figure 5 shows the average accuracy of the residual value stream architecture compared to the classic baseline, showing modest but sustained improvements over our single- and five-shot language understanding tasks. We observed a very high correlation between the two models' performance (0.993), which is expected for paired samples evolving over training steps with the same training data. We conducted a paired two-sample for means t-test on the average accuracy and found the p-values (both one-tail and two-tail) are much smaller than $\alpha = 0.05$ (the t-critical one- and two-tail values were 1.694 and 2.037, respectively). Therefore, we may reject the null hypothesis, *i.e.*, the mean difference between the paired samples is *not* zero. Given this, we confirm the improvement in performance at the 1B scale is statistically significant across these tasks.

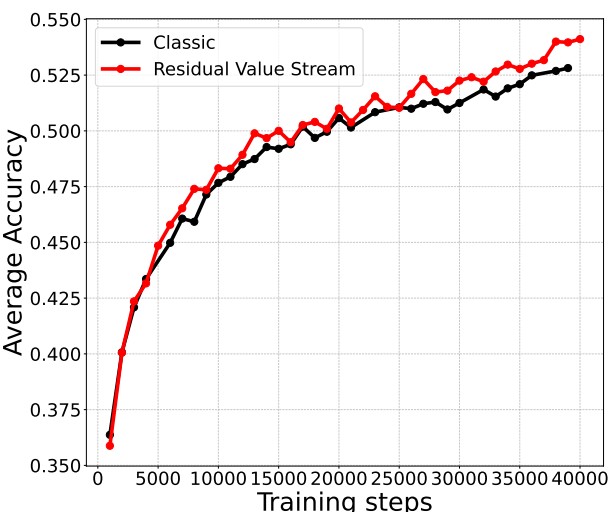

Figure 5: Average accuracy across training steps for our evaluations (single- and five-shot). Individual evaluation results can be found in Figure 14.

However, these gains appear localised to particular tasks. In Figure 14, we can see that the most noticeable gains are in the ARC and OpenBookQA tasks whereas there is almost no change in the PIQA and HellaSwag results. We therefore also evaluated our model on an extended version of the IOI task, which we dub *IOI-Hard*. In the IOI-Hard task, there are 25 sentences like:

- "{A} and {B} were commuting to the {PLACE}."

- "After lunch, {A} and {B} went to the {PLACE}."

- "The {PLACE} {A} and {B} went to had a {OBJECT}."

- "Friends {A} and {B} found a {OBJECT} at the {PLACE}."

- "{A} and {B} were planning to visit the {PLACE}."

where {A} and {B} are a unique pair of person names, {PLACE} is a unique physical place (*e.g.*, park, library, coffee shop, *etc.*), and {OBJECT} is a unique physical object (*e.g.*, book, pen, flower, key, *etc.*). Following 25 unique sentences of this type, there is a final, unfinished sentence designed to test if the model can correctly identify the indirect object. The final sentences randomly selected a pair of person names occurring in one of the previous 25 unique sentences, and matched the place and object (if mentioned) in one of the following final sentences:

- "At the {PLACE}, {GIVER} wanted to give a {OBJECT} to"

- "At the {PLACE}, {GIVER} decided to give the {OBJECT} to"

- "Then at the {PLACE}, {GIVER} gave a {OBJECT} to"

- "While at the {PLACE}, {GIVER} handed a {OBJECT} to"

where the {PLACE} and {OBJECT} was as before, the {GIVER} was randomly chosen as {A} or {B}, and the correct completion was {A} if {B} was chosen as the {GIVER} and vice-versa.

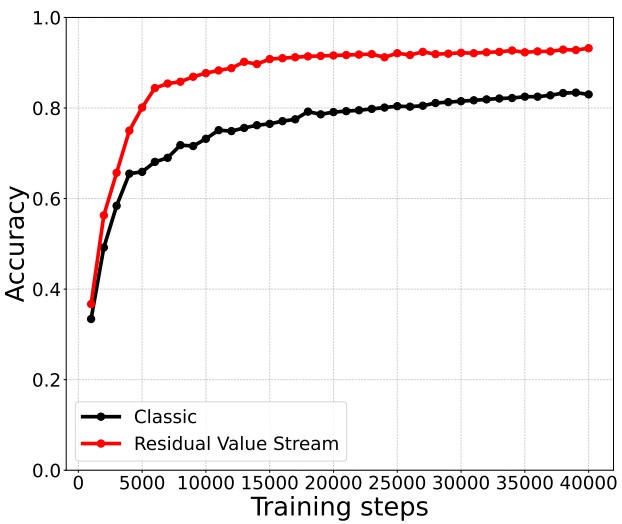

Figure 6: Accuracy across training steps for 1,000 completions of the IOI-Hard task evaluation.

Figure 6 shows the accuracy of the IOI-Hard task over training steps for both models, where 1,000 randomly-generated prompts of the above style were created, using 32 unique names, 30 unique objects, and 30 unique places. In this task, which is much more directly testing ICL ability, we see a very clear performance gain in our model compared to the classic model.

## 6 Discussion

Our study builds upon the understanding of ICL in Transformer models by exploring connections to associative memory models from computational neuroscience. We introduced AMICL, an associative memory model that performs ICL using an approach akin to a single-layer Transformer attention head but with an implied residual stream from the inputs to the values. This itself is notable given ICL is not typically seen in single-layer Transformers (Olsson et al., 2022) outside of simple linear regression tasks (Zhang et al., 2024a; Lu et al., 2024). Inspired by this, we proposed a novel residual stream architecture in Transformers, where information flows directly between attention heads. We demonstrate this in a two-layer Transformer, showing increased efficiency in learning on an ICL classification task. Moreover, by extrapolating this architecture to small Transformer-based language models, we illustrated enhanced ICL capabilities are possible on a larger and more naturalistic scale.

### 6.1 Connections to computational neuroscience

Our results offer potential insights for neural network design and our understanding of biological cognition:

- **Neural network architecture**. The simplicity of our neuroscience-inspired architectural modification, attention residual streams, provides a promising extension for designing more adaptive Transformer models. Given one can interpret this change as providing additional shared workspaces across model layers (in the form of the additional residual streams), our results can also be viewed as demonstrating another example where functional modularisation can provide a performance benefit (Kaiser & Hilgetag, 2006; Chen et al., 2013), which is also seen in cases of highly distributed representations (Voitov & Mrsic-Flogel, 2022), of which natural language appears no exception (Mikolov et al., 2013; Hernandez et al., 2024).

- **Biological cognition**. Drawing parallels between artificial networks and cognitive neuroscience theories contributes to a deeper understanding of memory systems in biological entities. As we have shown by our AMICL model, the associative memory framework is capable of ICL in a single layer, with an implied 'skip connection' present between the input and the values. This suggests that similar types of connections may analogously exist in biological networks to adapt and generalize from limited exposure. For instance, associative memory is often related to the hippocampal formation (Amit, 1989), an area in the brain important for memory and learning. An open question in this area of neuroscience is what the computational role of observed skip connections are. In particular, while some information in the hippocampus goes directly from area CA3 to CA1, another area – CA2 – is often skipped, but does receive input from CA3, which it then passes on to CA1 (Cui et al., 2013). This is interesting not just anatomically, but also functionally, since CA2 is considered responsible for specialised tasks and context-switching (Dudek et al., 2016; Robert et al., 2018). Whether our AMICL model or residual attention stream modification can be considered analogous to skip connections witnessed in the hippocampus remains to be studied. However, it offers a promising window, *e.g.*, one could now study the effects of changing the number and quality of residual attention streams, to test whether connection types or relative proportions similar to that seen in the hippocampus also show improved performance (for some types of data or tasks). We also suggest experimental neuroscientists perform a classical ablation study of CA3 to CA1 skip connections at different intensities and measure the performance of an object–pair recognition task with the same structure as described in §2.1 and §3.3. We hypothesise that the strength and number of remaining CA3 to CA1 skip connections will correlate with performance on the IC and IC2 tasks but not the IW task. Further, we also hypothesise the effect size of this performance difference will correlate with the effect size of the difference in performance caused by ablation of CA2 itself, since we theorise these CA3 to CA1 skip connections are mostly valuable only in the presence of CA2 to CA1 connections.

In establishing an additional potential mechanistic link between associative memory frameworks and ICL tasks, this work builds upon the broader neuroscience-inspired interpretation of Transformer attention mechanisms (Ramsauer et al., 2021). Indeed, Zhao (2023) conjectures that LLMs performing ICL do so using an associative memory model which performs a kind of pattern-completion using provided context tokens, conditioned by the learnt LLM parameters. Using this perspective, Zhao (2023) constructs different token sequences to be used as contexts to prepend onto the same final token sequence representing the tested task. By actively choosing context tokens which more 'closely' resemble the final tokens the LLM is being tested with, the LLM shows improved task performance, presumably by utilising the increased relevancy of the context tokens for ICL. This perspective is further developed in Jiang et al. (2024), who show how LLMs can be 'hijacked' by purposeful use of contexts with particular semantics. When LLMs such as Gemma-2B-IT and LLaMA-7B are given the context of "The Eiffel Tower is in the city of", they successfully predict the next token as " Paris". However, Jiang et al. (2024) demonstrate that prepending the context with the sentence "The Eiffel Tower is not in Chicago." a sufficient number of times, these LLMs incorrectly predict the next token as " Chicago". We may interpret these prepended data as acting (crudely) as 'distractors', and in the associative memory sense as causing an over-activation of competing memory items which interferes with accurate task performance.

Beyond associative memory, our architecture also shows structural similarities to other models from computational neuroscience. Notably, AMICL shares characteristics with successor representation (SR) models, particularly $\gamma$–models where a discount factor governs the contribution of future states (Janner et al., 2020).

In AMICL, the parameter $a$ plays a similar role, and in all of our implementations of the value residual stream is implicitly set to 1 (since we do not weight the added residual stream). Altering $a$ or weighting the value residual stream would enable control over the influence of past representations within the residual stream. This connection suggests that our model may be modified to encode a temporal structure over representations, akin to how SRs integrate over future expected states. Furthermore, recent work has highlighted parallels between SRs and temporal context models as applied to LLMs (Ji-An et al., 2024), particularly through the lens of reinforcement learning (RL) and temporal difference learning (Sutton, 1988; Gershman et al., 2012). These insights open a path for future work: instead of passing values from a single attention head, the model could be extended by learning weighted combinations of values across heads and layers – with the weights themselves learned via RL-based methods – further enriching the capacity to encode temporal or relational structure.

## 6.2 Potential representational benefit of residual value streams

The representational benefit of the residual attention stream modification, specifically the residual values stream, may be that it enhances the model's ability to retain and propagate contextual information across layers in a more persistent and structured way. In standard Transformers, attention values are recomputed independently at each layer, potentially discarding too much in the form of intermediate representations, which themselves retain relevant information regarding input associations. By directly passing the attention values from one layer to the next, the model effectively gains a memory-like mechanism that may help preserve previously computed associations, which could be particularly beneficial for tasks requiring ICL.

This residual pathway may enable a form of representational continuity: information that was relevant in earlier layers remains accessible in later layers, even if the current query-key similarity focuses elsewhere. This could allow the model to better align and integrate information across time or token positions, supporting pattern completion and associative recall, which can be interpreted as core functions of ICL. This may allow for more robust binding between stimulus features and outcomes, especially when only partial cues are available in context. Thus, the modification potentially enriches the model's internal representations with longer-range, more stable associations that are critical for adaptive, few-shot generalisation.

## 6.3 Limitations and future work

A key limitation of this paper is that, while it introduces a novel and neuroscience-inspired architectural modification and demonstrates improved ICL performance across synthetic tasks and a small-scale language model, its generalisability remains uncertain. The performance improvements, though statistically significant, are modest and task-specific when evaluated on broader benchmarks, such as PIQA and HellaSwag, where the proposed architecture shows minimal gains. Additionally, the residual stream mechanism is implemented in a relatively naïve fashion, leaving open questions about its optimal design and scalability to larger, more complex models. The paper also does not explore potential trade-offs, such as computational cost versus benefit or impacts on tasks unrelated to ICL, limiting conclusions about its broader utility in real-world applications.

Future efforts may seek to more deeply understand our results in the context of varying the structure and order of context tokens – in associative memory networks, and during inference and training of LLMs. Along these lines, Russin et al. (2024) recently showed that LLMs and Transformers exhibit similar improvements as seen in humans on tasks with and without rule-like structures, depending on the order and organisation of context and training tokens. In particular, ICL performance improved when context tokens appeared in semantically-relevant 'chunks' or 'blocks', as seen in humans when completing tasks with rule-like structures. Whereas, when training samples were interleaved, Transformers saw performance improvements (as measured by the extent to which the network rote-learnt training examples), similar to humans completing tasks lacking rule-like structures.

Other recent works have demonstrated how Transformer-based language models store and retrieve knowledge, using synthetic tasks to isolate specific mechanisms (Bietti et al., 2023; Nichani et al., 2024). Bietti et al. (2023) analyse how Transformers balance global knowledge learned during training with context-specific knowledge acquired during inference. They show that weight matrices function as associative memories,

with different learning dynamics for global bigrams and in-context bigrams, and provide theoretical insights into how gradients support this process. Nichani et al. (2024) focus on factual recall, proving that shallow Transformers can achieve near-optimal storage capacity using either self-attention or MLP components as associative memories. They also show that such models can trade off between these components and exhibit sequential learning dynamics during training. We believe it could by highly valuable for future work to analyse how such learning dynamics vary in the case of the residual values stream modification, and to explore whether an analogous modification could be made to the MLP components.

The performance improvements from introducing a residual values stream suggests that ICL can be thought of as an associative process where continuous information reinforcement enhances model memory, efficiency, and prediction accuracy on novel data. Nonetheless, this raises several questions. Firstly, more comprehensive testing on varied datasets and different scales of language models can address whether the improvements we observed generalise beyond our specific tasks and data setups. Further, it remains to be clarified whether similar benefits manifest when dealing with other natural language processing tasks, such as sentiment analysis or translation, or whether there exist any trade-offs between ICL and other abilities. Additionally, while these initial results present a compelling case for testing attention residual streams in larger models, further exploration of optimisation parameter settings for such architectures would strengthen understanding. Quantitative studies on computation cost versus accuracy improvements will also better-inform model architecture design and selection for deployment in real-world scenarios, as well as potential competition between learning flexibility and task accuracy.

### 6.4 Conclusion

This research demonstrates potential conceptual and practical advancements in enhancing Transformers' adaptive capabilities through a neuroscience-inspired mechanism. By bridging neural computational principles with associative memory insights, we offer new directions for research into more intelligent and dynamic models, with potential improvements for LLMs. Our associative memory model and Transformer architecture not only bolsters existing computational frameworks, but also offers fertile ground for computational neuroscientists to analyse the computational role of skip connections in memory systems. As we progress, these interdisciplinary approaches may ultimately yield richer cognitive models that parallel, or even emulate, biological intelligence.

### Acknowledgments

This effort was supported by the Cornell University SciAI Center. T.B. and C.E. were funded by the Office of Naval Research (ONR), under Grant Number N00014-23-1-2729. T.F. was supported by JSPS KAKENHI no. JP23H05476. T.B. thanks Gautam Reddy for code sharing, Björn Deiseroth for computing support, and Fabien C. Y. Benureau for helpful discussions.

### Code availability

Code used in this project can be found at `https://github.com/tfburns/AMICL-and-residual-attention-streams`.

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

## Appendix

## A Connection between associative memory models and modern neuroscience

Associative memory falls under the broader category of *long-term memory*—the type of memory capable of lasting a lifetime. Yet, memory also operates on at least two additional, functionally distinct timescales: *short-term memory* and *sensory memory*. Short-term (or working) memory retains information over tens of seconds and serves as a limited, passive buffer that enables manipulation and use of that information (Milner, 1955; Mongillo et al., 2008). In contrast, sensory memory operates on even briefer timescales – typically just a few seconds. It is the initial stage where sensory inputs such as visual (Sperling, 1963; Vogel et al., 2001), auditory (Darwin et al., 1972; Winkler & Cowan, 2005), and other sensory modalities (Gordon et al., 1993; Lederman & Klatzky, 2009) are actively "remembered" before either being discarded or encoded further. A general model for forming long-term memories thus involves: (1) receiving sensory input; (2) briefly storing features of that input in sensory memory; (3) selectively retaining and manipulating aspects of this input within short-term memory, possibly integrating multiple sources; and (4) consolidating the result into long-term memory. Theoretical and computational insights into these processes are valuable for two reasons: (i) they may inform the development of intelligent systems inspired by biological memory mechanisms; and (ii) they contribute to understanding the neural basis of memory, with potential applications in both clinical and cognitive contexts.

In both humans and other animals, associative memory typically refers to any form of long-term memory involving the linking or "association" of distinct stimuli, allowing the recall of one item upon encountering the other. Classical examples include name–face associations (Sperling et al., 2003), object–sound pairings (Preziosi & Coane, 2017), and object–location associations (Gilbert & Kesner, 2004). These forms are considered part of *explicit* or *declarative memory* (Ullman, 2004), which encompasses memories that can be consciously retrieved or verbalized. In contrast, *implicit* or *non-declarative memory* supports associative processes that occur unconsciously or automatically, such as those formed through classical conditioning (Maren, 2001; Christian & Thompson, 2003) and operant conditioning (Mackintosh, 1983; McSweeney & Murphy, 2014).

Computational models of implicit associative memory have a long history, beginning with foundational models such as the Rescorla–Wagner model (Rescorla & Wagner, 1972), which laid groundwork for the modern field of reinforcement learning (Daw & Doya, 2006). Explicit associative memory, on the other hand, appears to require greater computational sophistication, as suggested by the complexity of its biological underpinnings (Chaudhuri & Fiete, 2016; Clopath et al., 2017; Mau et al., 2020).

A seminal model of explicit associative memory is the associative memory model Marr (1971); Nakano (1972); Amari (1972); Little (1974), popularised as the Hopfield network (Hopfield, 1982). In its simplest form, when storing a single memory, the associative memory model allows for a configuration of neuron thresholds such that a partial activation reliably leads to full memory recall. This behaviour corresponds to attractor dynamics converging to a stable memory state, analogous to neuronal assemblies in the hippocampus (Wills et al., 2005; Pfeiffer & Foster, 2015; Rebola et al., 2017).

The hippocampus is widely regarded as a central structure in memory processing. Along with the entorhinal, perirhinal, and parahippocampal cortices, it plays a critical role in the formation of explicit memories (Scoville & Milner, 1957; Milner, 1966; Squire, 1992). It acts as a temporary store for new information, which is later consolidated in the cortex (Squire et al., 1989; Sutherland & Rudy, 1989). Traditionally, these processes have been attributed to Hebbian learning and long-term potentiation (Bliss & Gardner-Medwin, 1973; Gustafsson & Wigström, 1988), though recent research has highlighted additional mechanisms that may contribute to

memory formation and maintenance (Rigby et al., 2022; Groschner et al., 2022; Hedrick et al., 2022; Kato et al., 2022; Papadimitriou et al., 2020; Hart & Huk, 2020; Chipman et al., 2021; Perea et al., 2009; Doron et al., 2022; Etherington et al., 2010; Chavlis & Poirazi, 2021).

Despite its strengths, the classical associative memory model has limitations as a full account of long-term memory. For example, its memory capacity scales linearly with the number of neurons $n$, supporting approximately $0.14n$ stable patterns before performance degrades due to spurious attractors (Amit et al., 1985; McEliece et al., 1987; Bruck & Roychowdhury, 1990). This capacity is further reduced under correlated input conditions (Löwe, 1998), sparse connectivity (Treves & Amit, 1988; Löwe & Vermet, 2011), or both (Burns et al., 2022). Yet, biological systems often operate in such sparse regimes (Minai & Levy, 1993; Lansner, 2009; Barth & Poulet, 2012), and human memory regularly handles highly structured and overlapping information (Constantinescu et al., 2016; Aronov et al., 2017; Bellmund et al., 2018; Bao et al., 2019; Park et al., 2021; Griesbauer et al., 2022). Nevertheless, humans can remember thousands of highly similar images (Standing, 1973; Brady et al., 2008), recognize countless faces (Jenkins et al., 2018), learn tens of thousands of words (Brysbaert et al., 2016), and even recite over 100,000 digits of $\pi$ (Bellos, 2015), all without significant interference.

Although modern associative memory networks have achieved much higher theoretical capacities (Krotov & Hopfield, 2016; Demircigil et al., 2017), the empirical data and biological constraints – such as the finite number of neurons (Herculano-Houzel, 2009) and the metabolic costs of sustaining their connections (Bordone et al., 2019) – suggest there are deeper computational and physiological principles at work. Even within associative memory-type frameworks, memory capacity can be interpreted not just as storage volume, but as a trade-off involving recall fidelity, interference, and cognitive efficiency.

Still, these limitations do not diminish the relevance of associative memory networks or their successors for understanding memory phenomena in biological and machines. They remain powerful tools in neuroscience (Sathasivam & Wan Abdullah, 2008; Rizzuto & Kahana, 2001; Weber et al., 2017), machine learning (Widrich et al., 2020; Seidl et al., 2022), and in bridging the gap between artificial and biological systems (Sharma et al., 2022; Hoover et al., 2022; Chaudhuri & Fiete, 2019; Tyulmankov et al., 2021; Kozachkov et al., 2022). Substantial progress has already been made in expanding their capabilities, including increased capacity and efficiency (Storkey, 1997; Hopfieid, 2008; Krotov & Hopfield, 2016; Gripon & Berrou, 2011; Mofrad & Parker, 2017; Burns & Fukai, 2023), sparse representations (Kim et al., 2017; Hoffmann, 2019), and the incorporation of biologically inspired mechanisms (Watson et al., 2011a;b; Woodward et al., 2015; Burns et al., 2022; Burns, 2024). The current work aims to contribute further to these areas by: (1) understanding the capacity of the associative memory framework to perform ICL (demonstrated with the AMICL model); and (2) testing whether incorporating associative memory-inspired modifications to Transformers improves their capabilities (demonstrated with the residual values stream).

## B  Control experiments

In the following experiments, we use variants of a one-layer Transformer architecture and task as in §3. In all cases, the matrix $W_1^v$ is removed, to align with a naïve version of the AMICL model. The only difference in the following four cases is the number of ReLU layers and whether these are placed before (Figure 7), after (Figure 8), or both before and after (Figures 9 and 10) the single (value-absent) attention layer.

In all cases: the in weights (IW) task performs best, and close to perfect accuracy given sufficient training steps; the in-context (IC) task with novel data achieves approximately chance performance (20%); the in-context 2 (IC2) task with misaligned data initially performs near chance level but as the network begins memorising the training data (as illustrated through the delayed but superior improvement seen in the IW task), decays to 0% accuracy; and the test converges to a mid-point between the IW and IC task performance levels, indicating a near-equal performance trade-off occurs in the learnt algorithm.

Two other general trends are observable: (i) having ReLU layers follow the attention layer proves a smoother performance trajectory (compare Figures 7) and 8); and (ii) having more ReLU layers increases the speed of the learning trajectories.

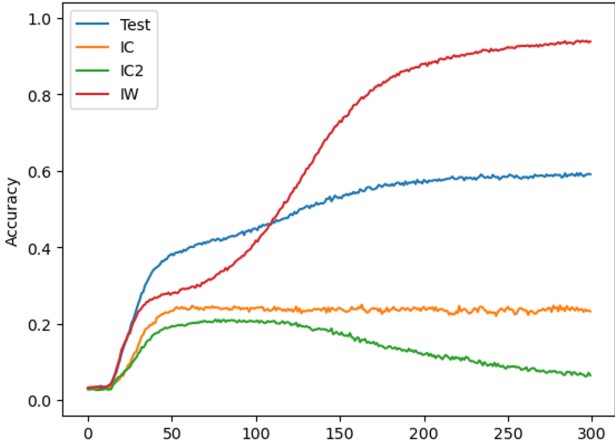

Figure 7: Performance of a single (value-absent) attention layer followed by three ReLU layers over training steps.

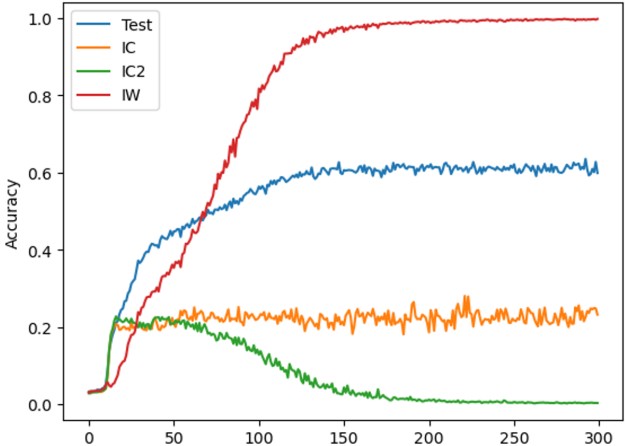

Figure 8: Performance of a model with three ReLU layers followed by a single (value-absent) attention layer over training steps.

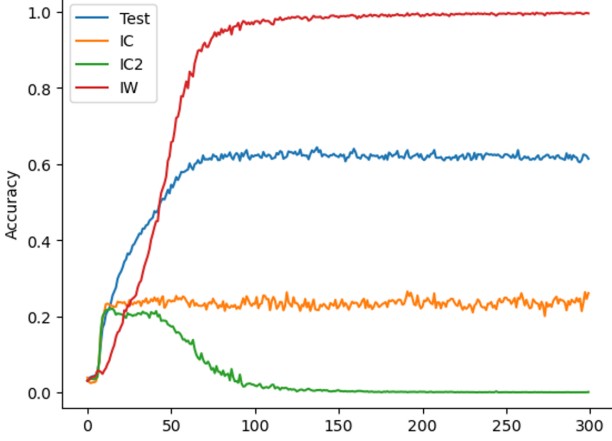

Figure 9: Performance of a model with two ReLU layers, followed by a single (value-absent) attention layer, followed by two ReLU layers over training steps.

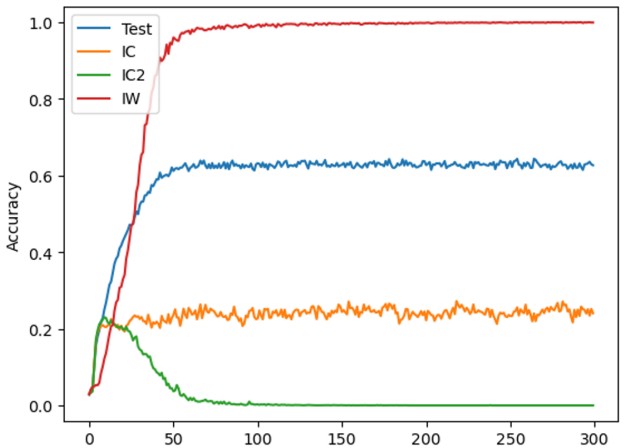

Figure 10: Performance of a model with three ReLU layers, followed by a single (value-absent) attention layer, followed by three ReLU layers over training steps.

## C    Extended figures

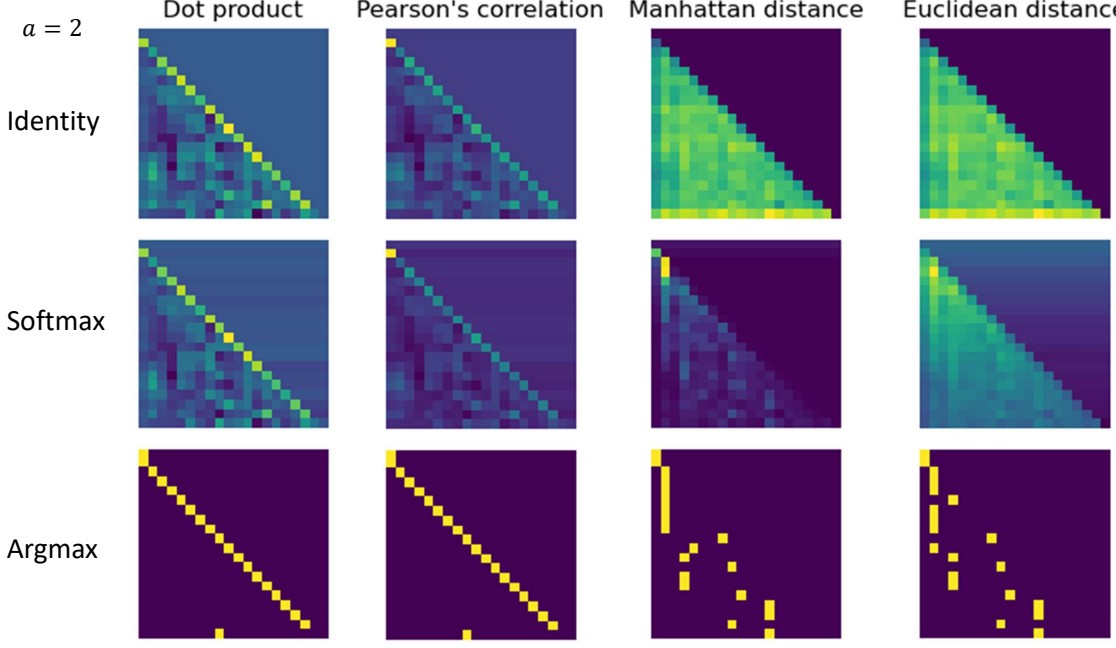

Figure 11: Attention matrices for label–object pairs in AMICL, using different similarity and separation functions with $a = 2$.

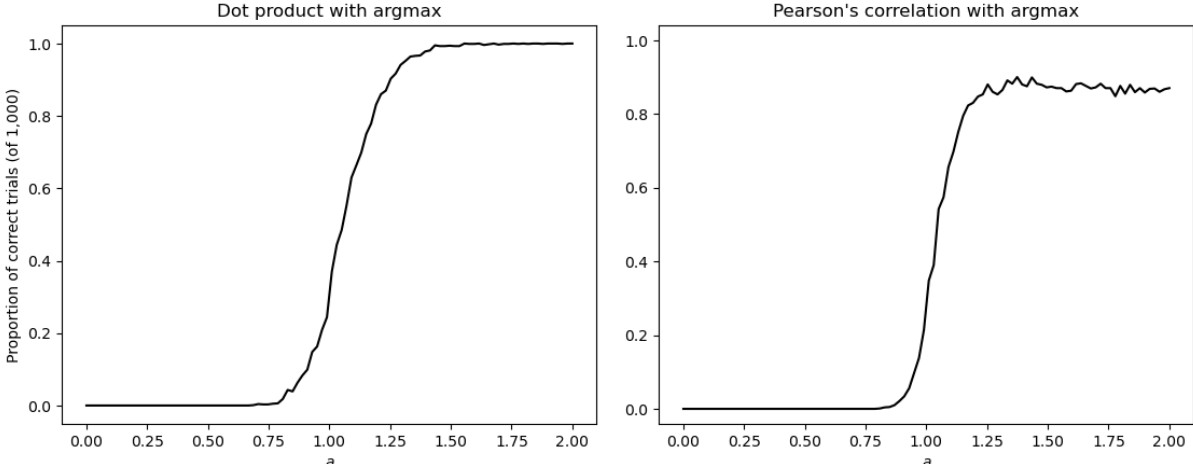

Figure 12: Proportion of correct trials, tested over $1,000$ trials for values between $a = 0$ and $a = 2$ in AMICL for label–object pairs using the DOT PRODUCT (left) and PEARSON'S CORRELATION (left) similarity with the ARGMAX separation function.

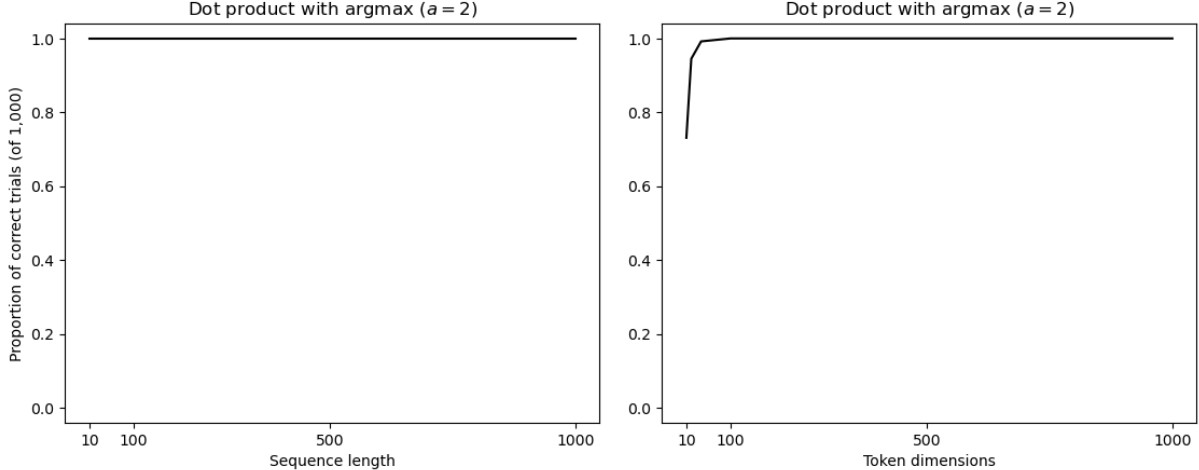

Figure 13: Proportion of correct trials, tested over $1,000$ trials for varying the values of the sequence length ($s$, left) and token dimensions ($e$, right) between 10 and $1,000$ in AMICL for label–object pairs using $a = 2$ with the DOT PRODUCT similarity function and ARGMAX separation function.

## D  Extended tables

Table 5: Mean ± standard deviation of training snapshot number where accuracy first exceeded 0.5 for the IC and IC2 tasks in the classic (unmodified), and residual queries, keys, and values stream networks.

|     | Classic | Queries (ours) | Keys (ours) | **Values (ours)** |
| --- | --- | --- | --- | --- |
| IC  | $51.0 \pm 4.24$ | $41.5 \pm 2.29$ | $49.75 \pm 0.43$ | $\mathbf{32.75} \pm 0.83$ |
| IC2 | $51.5 \pm 4.39$ | $41.5 \pm 1.66$ | $49.25 \pm 0.83$ | $\mathbf{33.25} \pm 1.09$ |

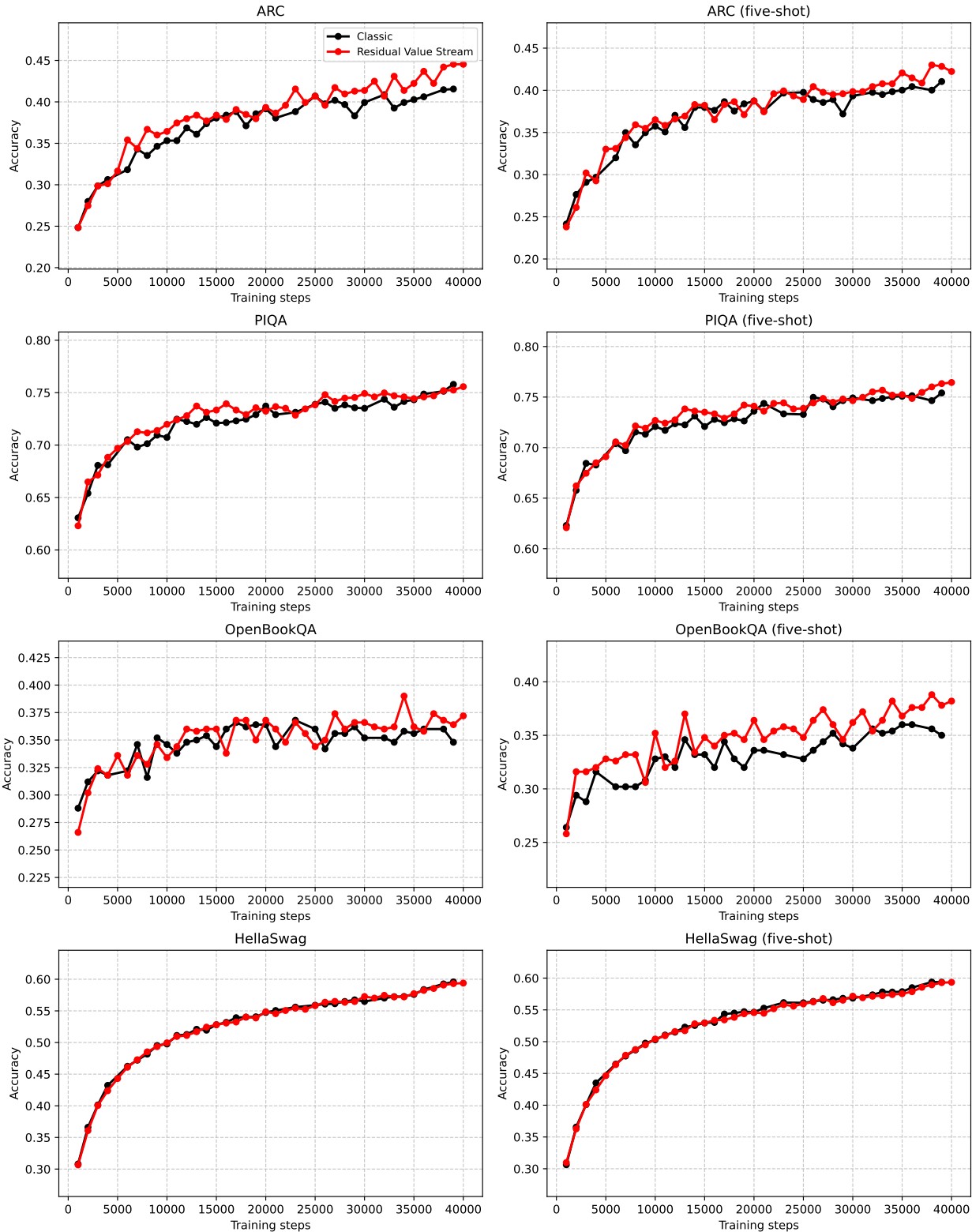

Figure 14: Accuracy across training steps for our evaluations of the 1B model, single- and five-shot.

Table 6: Mean ± standard deviation of training snapshot number where accuracy first exceeded 0.9 for the IC and IC2 tasks in the classic (unmodified), and residual queries, keys, and values stream networks.

|      | Classic          | Queries (ours)   | Keys (ours)      | **Values (ours)**   |
|------|------------------|------------------|------------------|---------------------|
| IC   | 59.25 ± 4.49     | 48.25 ± 1.48     | 57.5 ± 0.5       | **42.25** ± 1.3     |
| IC2  | 59.0 ± 4.53      | 48.0 ± 1.87      | 57.25 ± 0.83     | **42.0** ± 0.71     |

## E   Notation and abbreviations tables

A comprehensive list of all notations and abbreviations used in this paper is provided in the tables below.

**Abbreviations**

| | |
|---|---|
| LLMs | large language models |
| ICL | in-context learning |
| LMs | langauge models |
| i.i.d. | independent and identically distributed |
| IW | in-weight |
| IC | in-context |
| IC2 | in-context 2 |
| IOI | indirect object identification |
| CA1, CA2, CA3 | cornu Ammonis 1, 2, 3 |

**Variables**

| | |
|---|---|
| $X_i$ | The input sequence data $X_i \in \mathbb{R}^{e \times s}$, with $s$ column vectors, corresponding to the token embeddings, each of dimension $e$. Where the subscript $i$ is present, this denotes the data is taken from the $i$–th Transformer layer, *i.e.*, the residual stream data. |
| $x_i$ | A token embedding $x_i \in \mathbb{R}^e$, where the subscript $i$ denotes the column position in the input sequence $X$. |
| $x_s$ | The token embedding $x_i \in \mathbb{R}^e$ of the final column, *i.e.*, the final token, in the input sequence $X$. |
| $o^i$ | An object token embedding $o^i \in \mathbb{R}^e$, where the superscript $i$ identifies the object–label identity. |
| $l^i$ | A label token embedding $l^i \in \mathbb{R}^e$, where the superscript $i$ identifies the object–label identity. |
| $\mu_i$ | A vector $\mu_i \in \mathbb{R}^e$ whose components are i.i.d. sampled from a normal distribution having mean zero and variance $1/e$. Used for constructing the token embeddings of object or label $i$, where for each object, $o^i$, and label, $l^i$, the vector $\mu_i$ is fixed. |
| $\varepsilon$ | A fixed real number $\varepsilon \in \mathbb{R}$ which controls the inter-instance variability of objects, and is set to 0.1 unless stated otherwise. |
| $\eta$ | A vector $\eta \in \mathbb{R}^e$ whose components are i.i.d. sampled from a normal distribution having mean zero and variance $1/e$. Redrawn and used for adding inter-instance variability in the construction of each object token embedding. |
| $W_i^q, W_i^k$ | Query and key weight matrices $W^q, W^k \in \mathbb{R}^{\hbar \times e}$, respectively, of the $i$–th Transformer layer. |
| $W_i^v$ | Value weight matrix $W^v \in \mathbb{R}^{v \times e}$ of the $i$–th Transformer layer. |
| $Q_i$ | Queries matrix $Q \in \mathbb{R}^{\hbar \times s}$ of the $i$–th Transformer layer, calculated using $W_i^q$ and $X_i$. |
| $K_i$ | Keys matrix $K \in \mathbb{R}^{\hbar \times s}$ of the $i$–th Transformer layer, calculated using $W_i^k$ and $X_i$. |
| $V_i$ | Values matrix $V \in \mathbb{R}^{v \times s}$ of the $i$–th Transformer layer, calculated using $W_i^v$ and $X_i$. |
| $S_i$ | The scores matrix, $S_i \in \mathbb{R}^{s \times s}$, which is equal to $K_i^T Q_i$. |
| $q_i$ | A queries column vector, $q_i \in \mathbb{R}^{\hbar}$, where the subscript $i$ denotes the column position in the queries matrix $Q$. |
| $k_i$ | A keys column vector, $k_i \in \mathbb{R}^{\hbar}$, where the subscript $i$ denotes the column position in the keys matrix $K$. |
| $v_i$ | A values column vector, $v_i \in \mathbb{R}^{v}$, where the subscript $i$ denotes the column position in the values matrix $V$. |

**Dimensions**

| | |
|---|---|
| $e$ | Dimensionality $e \in \mathbb{N}^+$ of each token embedding. |
| $s$ | Number of tokens $s \in \mathbb{N}^+$ in the input sequence data $X$. |
| $\hbar$ | Reduced token embedding dimension $\hbar \in \mathbb{N}^+$ for keys and queries in the attention operation. |
| $v$ | Reduced token embedding dimension $v \in \mathbb{N}^+$ for values in the attention operation. |
| $\ell$ | Number of unique labels $\ell \in \mathbb{N}^+$ in the input sequence data $X$. |

