# OpenReview forum: "Associative memory inspires improvements for in-context learning using a novel attention residual stream architecture"
_TMLR — Accepted by TMLR_

### Review · Reviewer_rkAz · 2025-04-08

**Summary Of Contributions:**

This short paper brings an associative memory perspective to bear on the phenomenon of in-context learning in Transformers. Its main contribution is a proposed augmentation of the standard Transformer architecture by additional residual streams carrying value information between attention blocks.

**Audience:**

Yes

**Claims And Evidence:**

Yes

**Requested Changes:**

I have already discussed at some length my concerns regarding the manuscript's clarity and its biological claims under Strengths and Weaknesses. In short, I think substantial re-writing would be required in order for me to recommend acceptance. Here I list other concerns.

- In the discussion, the authors state that "This itself is notable given ICL is not typically seen in single-layer Transformers (Olsson et al., 2022)." This claim must be qualified, as it depends on what ICL task one has in mind. As shown by Zhang et al., "Trained Transformers Learn Linear Models In-Context" and Lu et al., "Asymptotic theory of in-context learning by linear attention", a single attention block is sufficient for ICL of simple linear regression tasks.

- References to a line of work by Bietti and colleagues analyzing Transformers from an associative memory perspective (e.g., "Birth of a Transformer: A Memory Viewpoint" or "Understanding Factual Recall in Transformers via Associative Memories") are notably absent.

- One other limitation of this work in its current form is that it uses only two datasets: the synthetic ICL classification task from Reddy (2024), and the TinyStories dataset. It would be interesting to probe whether the proposed model displays strong performance on other synthetic ICL tasks, e.g., in-context regression.

**Strengths And Weaknesses:**

The main weakness of this manuscript is that it makes a number of imprecise and poorly-justified claims. I think the main idea is somewhat interesting, but the clarity and precision of the paper must be improved.

First, I think the "AMICL" model could be introduced more clearly. This is extremely important, as this is the key conceptual contribution of the paper. I suggest that the authors unpack the last few paragraphs of Section 2; for instance they might convert the equations for the keys and queries into display equations. There are also a number of statements regarding the construction of the model that require justification, for instance:

- "The value of a can be any arbitrary positive real value, but, after testing within the range [0, 2], is set as a = 2." Where is justification for this provided? Why is an upper limit of 2 selected?

- "Seen through the lens of the traditional self-attention mechanism, construction of the AMICL model in Section 2 can be interpreted as implying that creating the values V in a way which more explicitly retains the prior input X can facilitate ICL." How does the construction of the model imply this, until it is demonstrated that this model displays advantages in ICL performance?

I also have strong concerns regarding the authors' claims about their model's relationship to biology. Most simply, the authors claim that their residual value stream architecture is "biologically-inspired". The study of associative memory models has its roots in neuroscience, but it is not clear to me that there is anything inherently biological about this architectural modification. I list several more concerns regarding biological claims below:

- In the Introduction, the authors state that "Developing models which connect our understanding of common mechanisms underlying these related phenomena in both artificial and natural intelligence may provide valuable insights for developing more adaptive and versatile language models, in addition to helping us more deeply understand the brain." The assumption that there are necessarily common mechanisms is, in my opinion, quite non-trivial. To wit, showing that common mechanisms underlie rapid learning in LLMs and in animals would be a highly significant discovery.

- The Discussion includes two paragraphs on potential biological insights arising from the proposed models, which make sweeping claims without adequate justification. First, how can the results "be viewed as supporting cognitive functional specialisation and modularisation as being both natural and commonplace in neural systems"? Perhaps they demonstrate one example where functional modularization can provide a performance benefit, but how does this show these features to be "natural and commonplace"? I do not follow. Second, the authors raise the question of whether their "AMICL" model or residual attention stream modification can be considered analogous to skip connections witnessed in the hippocampus". However, they provide no specific predictions that could be tested, merely an analogy at the level of the existence of skip connections. I would suggest that the authors reify or remove these claims.

---

> ### Author Response · Authors · 2025-06-03
> **Response**
>
> We are grateful for Reviewer rkAz’s comprehensive review, including many specific and actionable suggestions to improve the paper.
>
> To address the weaknesses raised, in order of mention in Reviewer rkAz’s review –
>
> *Regarding the AMICL model introduction and how it inspires the residual values stream modification:*
>
> 0. We unpacked and expanded on explanations in what is now §2.2 to more comprehensively explain the AMICL model and provide better guidance for understanding the visual depiction of the model in Figure 1
> 1. We explained the justifications for analysing $a$ in the range of $[0,2]$ and ultimately setting it to $a=2$ in the final analyses in §2.2
> 2. We explained that the reason for choosing the attention values to have the identity of the original data in the AMICL model is because we cast the ICL task as an auto-association problem, and by the universal associative memory framework of Millidge et al. (2022), the projection function in the case of auto-association is the identity function (§2.2); then, via the same framework and by the same rationale, this led us to consider adding tying the values more closely to the input data (while still retaining the expressive power of the self-attention mechanism; see §3.1).
>
> *Regarding the connection of our models to neuroscience or biology:*
>
> 0. We think the phrase “biologically-inspired” may have been misleading, too strong, and/or too broad. Instead, we have replaced it with the phrase “neuroscience-inspired” and attempted to limit our claim of direct biological relevance generally.
> 1. To clarify and limit our general claim regarding the connection between associative memory models and modern neuroscience, we have added a new appendix titled “Connection between associative memory models and modern neuroscience”, which concludes with the following description of our contribution to this connection: “(1) understanding the capacity of the associative memory framework to perform ICL (demonstrated with the AMICL model); and (2) testing whether incorporating associative memory-inspired modifications to Transformers improves their capabilities (demonstrated with the residual values stream).”
> 2. (a) We have reduced the claim regarding functional modularisation to what was suggested: “[demonstrates an] example where functional modularization can provide a performance benefit”.
> (b) As suggested, we have specified an experiment and testable hypothesis relating the connection between our models and skip connections in the hippocampus.
>
> To address the directly requested changes:
> 1. We have qualified the single-layer Transformers ICL capability claim with the mention that simple linear regression tasks have been shown to be possible with single-layers, and cited the two papers mentioned (Zhang et al. and Lu et al.).
> 2. We have added references to and discussed the two identified papers by Bietti and colleagues in §6.2.
> 3. During the last two weeks, we prioritised our engineering time and compute budget on training and evaluating two 1B language models. Nevertheless, we believe it is indeed worthwhile to try the additional synthetic task of in-context regression, probably at the smaller scales. Reviewer KET7 also suggested something similar. We therefore propose to complete these in the next two weeks and/or as part of the “minor revisions” phase if Reviewers rkAz and KET7, as well as the Action Editor, agrees.
>
> Reviewer rkAz provided us with a very diligent review, which very carefully analysed our claims and helped us improve the paper; thank you very much for raising these points and suggestions, which we believe make the paper much clearer and stronger.

---

> > ### Comment · Reviewer_rkAz · 2025-06-06
> >
> > Thank you for your response. The revised manuscript does clarify the presentation of the AMICL model, and addresses some of my concerns regarding the justification provided for design choices. I have a few remaining questions and comments:
> >
> > - The phrase "connect our understanding of common mechanisms underlying these related phenomena" persists in the revised manuscript, and I continue to think that it should be dropped.
> >
> > - Figures 5 and 9 do not convince me that the improvement in performance is statistically significant. I second Reviewer g8Zb's concerns regarding the potential infeasibility of evaluating whether performance improvements at scale.
> >
> > - What is your goal in mentioning the successor representation? The link here is not clear to me.

---

> > > ### Author Response · Authors · 2025-06-12
> > > **Response 2**
> > >
> > > Thank you for your response. We are glad you found our first revision helpful in addressing some of your concerns.
> > >
> > > To address your remaining questions and comments:
> > >
> > > 1. We have replaced the sentence containing the previous phrase with "Developing models rooted in computational neuroscience, such as associative memory, to connect our understanding of related phenomena in both artificial and natural intelligence may provide valuable insights for developing more adaptive and versatile language models, in addition to helping us more deeply understand more general neural systems (see Appendix A for discussion on the connection between associative memory models and modern neuroscience)." What we mean to refer to here is the long-running and modern body of work of associative memory in this area. If this phrase is still problematic, however, we would welcome a direct suggestion on a better phrase to capture our intended meaning.
> > >
> > > 2. At the time of writing this reply, Reviewer g8Zb's stated concern regarding whether the performance improvements scale was written before the 1B results were added and does not yet appear responsive to these new results. We agree, however, that the performance improvement looks modest. Indeed, we observed a very high correlation between the two models' performance, which is expected for paired samples evolving over training steps with the same training data. However, we conducted a paired two-sample for means t-test on the average accuracy and found the p-values (both one-tail and two-tail) are much smaller than $\alpha=0.05$ (the t-critical one-tail value was $1.694$ and the t-critical two-tail value was $2.037$), which means we may reject the null hypothesis, i.e., that we may say the mean difference between the paired samples is *not* zero. Given this, we confirm the improvement in performance at the 1B scale is statistically significant across these tasks. Nevertheless, this performance improvement is still modest and localised to particular tasks. We therefore additionally introduce and evaluate our models on the IOI-Hard task, which is an extension of the IOI task we tested with the smaller language model. This task much more directly tests ICL ability, and we see a very clear performance gain in our model compared to the classic model.
> > >
> > > 3. Thank you for the chance to clarify this point. We agree that the connection to successor representations was not clearly stated in the original text. We have revised the discussion to clarify that we are drawing an analogy between the way AMICL integrates past representations via the parameter $a$ (which is implicitly set to $1$ in all of our implementations of the value residual stream, since we do not weight the added residual stream), and the way successor representations accumulate future state expectations using a discount factor $\gamma$. This connection is speculative but meaningful: it frames AMICL as implicitly encoding a temporal structure over inputs, akin to how SRs capture predictive relationships. We also now clarify how this perspective may guide future architectural variants using learned weighting schemes inspired by reinforcement learning.

---

> > > > ### Comment · Reviewer_rkAz · 2025-06-17
> > > >
> > > > Thank you for your response. I think the revised manuscript adequately substantiates its claims and acknowledges its limitations, so I've updated my assessment of whether it satisfies TMLR's "claims and evidence" criterion. I think it is suitable for publication in its current form.

---

### Review · Reviewer_g8Zb · 2025-04-14

**Summary Of Contributions:**

This paper proposed value stream as a residual branch to enhance the in-context learning capabilities of large language models. The evaluation is in the ICL-specific dataset. In a two-layer Transformer, value stream shows better convergence speed than baseline models. In 8-layer small language models, value stream shows a better performance on ICL tasks.

**Audience:**

No

**Broader Impact Concerns:**

No other concerns.

**Claims And Evidence:**

No

**Requested Changes:**

Personally, I completely understand that the small-scale of the experiment is due to the limited computation resources available. However, I do not believe that the experiments on the very small model can make any effective evaluations and conclusions.  I think the authors are better to try other research directions.

If focused on the the paper itself, I think the authors should either make the theoretical or empirical studies more solid. For theory perspective, the paper should prove that value residual broadens the expressiveness of the baseline Transformers. For empirical perspective, the real ICL and general evaluation tasks should be included. Adding the scaling analysis is better since the advantages of a simple residual enhancement can easily disappear with the scaling model depth.

**Strengths And Weaknesses:**

Strengths:
1. The definition of in-context learning is quite clear. The further analysis and case studies can represent the real ICL behaviors of language models.
2. The experiment setting is computation-friendly. Using a two-layer Transformer for toy-dataset experiments are cheap and easy to reproduce.

Weaknesses:
1. Most of the experiments con conducted under a very small scale. Usually, the toy dataset shows the intrinsic capability of an architecture, where the accuracy is near to 100% or random. However, The experiments in Figure 3 only shows the convergence speed, which can not show the capability of difference architectures. For example, [1] shows that data-driven priors make big influence on the toy-dataset performance.
2. Evaluation on real in-context training tasks is infeasible for only 8-million model size. Therefore, the experiment is not solid for the claim that value stream improves ICL.
3. The value is only the linear projection of input layers. Therefore, most of the properties in the value residual can be obtained from hidden residual:
$$
V_2=W_2^V \mathrm{norm}(X_1+X_2)+V_1=W_2^V\mathrm{norm}(X_1+X_2)+W_1^V\mathrm{norm}(X_1)
$$
Since$X_1+X_2$ already contains the residual information of $X_1$, another residual can not probably bring new capabilities.


[1] Never Train from Scratch: FAIR COMPARISON OF LONGSEQUENCE MODELS REQUIRES DATA-DRIVEN PRIORS

---

> ### Author Response · Authors · 2025-06-03
> **Response**
>
> We highly appreciate Reviewer g8Zb’s frank feedback, but also their understanding regarding our computational limitations. We were, however, inspired by this feedback, and have very fortunately been granted some limited compute access for us to perform the larger experiments which are necessary to validate our idea. Over the past two weeks, we have dedicated all of our engineering time and compute resources to this effort.
>
> So, in response to the identified weaknesses and corresponding requested changes, we opted for what we hope will lead to a more impactful result, which is empirical; in particular, we:
> - trained two 1B Llama-style language models on 84B English-language tokens from the LLM pre-training Nemotron-CC-HQ dataset (Su et al., 2024) – one with and one without our attention residual stream modification
> - after training these models, we evaluated them on single- and five-shot natural language understanding tasks, where the models do not (and are not expected to) reach 100% accuracy due to the difficulty of the tasks compared to the size of the models
> - although more modest, our results show there is a sustained improvement at the 1B scale
>
> These additional experiments and other changes requested by different reviewers, have been incorporated into the revised paper.
>
> We are once again very thankful for Reviewer g8Zb’s expert advice.

---

### Review · Reviewer_KET7 · 2025-05-18

**Summary Of Contributions:**

This work explores the relationship between computational neuroscience approaches to associative memory (e.g., the literature on hopfield networks) and in-context learning in transformers. A variant on the self-attention mechanism is shown to be capable of in-context learning, and the relationship to associative memory is analyzed. Taking inspiration from this, the authors then propose an architectural modification to transformers, and present evidence that this modification improves performance in both a toy in-context learning task and naturalistic pretraining.

**Audience:**

Yes

**Broader Impact Concerns:**

There is no need for a broader impact statement.

**Claims And Evidence:**

Yes

**Requested Changes:**

I think the paper would benefit from addressing the following issues:
- A tighter connection should be established between the AMICL model and the proposed modification to transformers. Additionally, other models could be explored that have a stronger connection to the AMICL model, e.g., eliminating the $W^v$ matrices, or incorporating residual stream connections between the input in layer l-1 and the value in layer l.
- The decision to make values equal to inputs in the AMICL model should be better motivated, especially given that this is the main inspiration for the modified transformer architecture. What is the basis for this design choice? Is it based on a particular source of biological inspiration, or is there a specific reason to think that it should yield improved performance?
- More explanation should be provided for why the modified transformer architecture yields performance benefits on the IOI task, despite not meaningfully improving overall language modeling performance.
- More control models should be explored, especially including other possible configurations for additional residual streams.
- In figure 3, the bottom right plot shows loss increasing for the IC2 task. This seems inconsistent with the increasing accuracy for this task. Is there an explanation for this discrepancy?

**Strengths And Weaknesses:**

In general, I think the connection between computational neuroscience models of associative memory and in-context learning in transformers is interesting and deserving of further study. Additionally, this paper has a number of interesting findings, and the proposed architectural modification seems to produce some importance benefits.

The primary weakness is that, in its current form, many aspects of the paper seem somewhat disjointed or not sufficiently motivated. For example, the proposed architectural modification -- a residual stream connecting value embeddings between successive layers -- seems reasonable, and produces some benefits, but the connection to the initial associative memory experiments is tenuous. The modification is motivated by noting that in the initial AMICL model, values are simply a copy of the input token embeddings. However, this does not directly translate into the proposed modification. The proposed modification provides a direct path between value embeddings in different layers, not between the input and the value embeddings. A closer match to the AMICL model would be simply eliminate the value embedding matrices ($W^v$), or to have a residual stream from the inputs to value embeddings at downstream layers.

Another concern is that is that there's no justification provided for this initial decision (making values equal to inputs) in the AMICL model. This appears to be an arbitrary design decision in the AMICL model, rather than being based either on a specific source of biological inspiration, or factors related to performance of the model. So, although there is some connection between this design choice in the AMICL model and the proposed modification to transformers, it does not seem to have anything to do with associative memory per se. This somewhat undermines the framing of the paper, according to which the modification to the transformer architecture is supposed to be related to or inspired by associative memory models in computational neuroscience.

Additionally, although the proposed modification does seem to yield performance benefits on the IOI task, there is not much explanation or discussion of *why* this modification yields improved performance on this particular task, especially given that the modified architecture doesn't really yield improved overall performance on the language modeling task used for training.

Finally, it might be informative to include additional control models in the experiments. In particular, I am curious whether it matters that the additional residual streams are passed between corresponding components (e.g. from values in layer l-1 to values in layer l), or do the performance benefits come merely from having additional pathways to aid in backpropagating gradients? For instance, it would be interesting to evaluate a version of the model with residual streams between queries/keys in layer l-1 and values in layer l, or simply randomly adding residual streams between elements in subsequent layers.

---

> ### Author Response · Authors · 2025-06-03
> **Response**
>
> We thank Reviewer KET7 for sharing extensive and valuable comments on our paper.
>
> In response to the requested changes to improve our work, we:
> 1. Developed a tighter connection between the AMICL model and the residual value stream (§3.1).
> 2. Explained that the reason for choosing the values to have the identity of the original data is because we cast the ICL task as an auto-association problem, and by the universal associative memory framework of Millidge et al. (2022), the projection function in the case of auto-association is the identity function (§2.2).
> 3. Provided explanations as for why the residual values stream variant outperforms the classic model so handedly in the IOI task despite the two models having very similar final loss values and the former having only a slightly smaller loss (end of §4.2).
> 4. During the last two weeks, we prioritised our engineering time and compute budget on training and evaluating two 1B language models. Nevertheless, we believe it is indeed worthwhile to add some additional controls, probably at the smaller scales (including other possible configurations for additional residual streams, as Reviewer KET7 suggests). Reviewer rkAz also suggested something similar. We therefore propose to complete these in the next two weeks and/or as part of the “minor revisions” phase if Reviewers KET7 and rkAz, as well as the Action Editor, agrees.
> 5. Regarding why the loss apparently increased on the IC2 task: this was a numerical error and is now corrected in the updated manuscript. Thank you for alerting us to this. For your information, it was due to the labels being (incorrectly) unchanged in the loss function implementation for that task.
>
> Regarding the primary weakness identified by Reviewer KET7: we agree that in our original submission, we did not do a good job of motivating and explaining the AMICL model, the residual value stream, and their connections to associative memory. We hope that our revisions mentioned in points 1 and 2 above (§3.1 and §2.2 of the text, respectively) help clarify this, and if we can make this even clearer we would value your suggestions on further edits or additions. The core part that was probably insufficiently described previously was the casting of the ICL task as an auto-association task, which in the description of the universal associative memory framework of Millidge et al. (2022) makes the values taking the identity of the input data natural. Further, although we did initially try experiments such as eliminating the $W^v$ matrix, this did not lead to good results and theoretically limits the expressibility of the model (also see Burns (2024)).
>
> We greatly appreciate Reviewer 4pDz for providing their detailed review of our paper. We strongly agree that the connection between computational neuroscience models of associative memory and in-context learning in Transformers is interesting and deserving of further study. We are therefore happy that we have been able to share a number of interesting findings, including the proposed architectural modification and its benefits (now including up to the scale of a 1B Llama-like language model trained on 84B tokens).

---

> > ### Comment · Reviewer_KET7 · 2025-06-18
> >
> > Thank you to the authors for these replies. I have some followup comments below:
> >
> > - The revised manuscript justifies the decision to use an identity function for the values by appealing to a previous paper (Millidge et al., 2022), but this still appears to be a somewhat arbitrary decision. There is no stated normative justification for this design decision, or justification in terms of biological inspiration.
> > - I have read the revised section 3.1, but it is still unclear to me how the residual stream between values is related to the use of an identity function for values in the AMICL model. The authors state that they tried using an identity function for values in the transformer model, but this resulted in lower performance. This suggests that setting values to be identical to inputs is not an effective architectural design choice.
> > - Without additional control experiments, it is unclear why the modified architecture improves performance, i.e. whether it is specific to the modified values, or whether it is more generally a benefit of additional residual connections.
> >
> > Overall, I am still left with the impression that the associative memory model and the modified transformer architecture are not really very closely related, and that more control experiments are needed to understand why the modified architecture is helpful.

---

### Review · Reviewer_4pDz · 2025-05-19

**Summary Of Contributions:**

The paper introduces two main contributions:
- Associative Memory for In-Context Learning (AMICL):
A one-layer associative memory model inspired by biological memory systems that can perform in-context learning (ICL) using a simplified attention-like mechanism. It demonstrates that even a single-layer architecture, when designed with biologically-inspired mechanisms, can support effective ICL.
- Residual Attention Stream Architecture:
A architectural modification to Transformers that introduces residual connections between the value vectors of attention heads across layers. This residual stream improves the speed and accuracy of ICL during training, especially in small models, and scales well to more realistic language tasks (e.g., TinyStories, IOI tasks).

**Audience:**

Yes

**Claims And Evidence:**

No

**Requested Changes:**

- Reconstruct the layout, add more subsections to better demonstrate the paper.
- Train on larger language models, at least show some scaling curves.
- Select a harder dataset where the models cannot reach 100% accuracy to evaluate.

**Strengths And Weaknesses:**

Strengths:
- The paper bridges neuroscience and machine learning by drawing on associative memory models from biological systems to improve Transformer architectures. This cross-disciplinary approach is novel.

Weaknesses:
- The writing needs to be completely polished. There are no subsections in method part and experiment part at all, making it difficult for readers to follow. There should be reasonable subsections where you introduce background, settings, etc., instead of mixing everything together.
- The experiments are performed on language models that are too small. Considering the size of current LLMs, the authors need to conduct experiments on few hundred million parameter models to be more persuasive.
- Comparing the steps to reach 100% accuracy as shown in Figure 3 is not a good metric. The authors should pick a hard enough dataset and compare accuracy under the same steps. This is more consistent with the actual situation.
- For the sixth sub-figure in the figure 3, why is the loss increasing on IC2 task?

---

> ### Author Response · Authors · 2025-06-03
> **Response**
>
> We thank Reviewer 4pDz for their thoughtful review and providing clear guidance on ways to improve our paper.
>
> In response to the identified weaknesses in the original submission and corresponding requested changes, we have:
> - polished the writing throughout and added more subsections to improve readability
> - trained 1B Llama-style language models on 84B English-language tokens from a general LLM pre-training dataset with and without the attention residual stream
> - we evaluated the 1B language models on natural language question understanding tasks, where the models do not (and are not expected to) reach 100% accuracy due to the difficulty of the tasks compared to the size of the models
>
> Regarding the question for why, in the sixth sub-figure of Figure 3, the loss increases on the IC2 task: thanks for spotting this! This was a numerical error due to the labels being unchanged in the loss function. It is now corrected in the updated manuscript.
>
> We once again wish to thank Reviewer 4pDz for their support in improving our paper. We are also thankful they appreciated our novel and cross-disciplinary approach in this work, bridging concepts from neuroscience and machine learning to improve Transformer architectures.

---

### Decision · Action_Editor_i3q3 · 2025-06-19

**Recommendation:** Accept with minor revision

**Additional Comments:**

- Explicitly address the remaining critiques from the reviewers in the "discussion" section or if they are not addressable, then highlight in the "limitations" section.
- Run some additional tasks isolating the benefits of the proposed architectural modification in the context of ICL (which the authors have claimed that they have done already but not added to the manuscript).
- Explain the representational benefit of the modification a bit more clearly.
- Ensure the presentation of the experimental results is clear, and the statistical significance is adequately explained.

**Audience:**

Yes

**Audience Explanation:**

ICL is a topic of interest to the broader machine learning community and this paper connects neuroscience and ICL so would have interest from a decent audience. All reviewers agree with this.

**Claims And Evidence:**

Yes

**Claims Explanation:**

The paper introduces an attention residual stream architecture for Transformers, inspired by associative memory models from computational neuroscience, which allows information to flow directly between the value vectors of attention heads in successive layers. The authors claim that this modification enhances a model's ICL capabilities through experimentation on small-scale and some large-scale settings.

The reviewers raised several issues with the original submission:
- Multiple reviewers (4pDz, g8Zb) thought that the experiments were conducted on models that were too small to convincingly demonstrate the scalability or relevance of the proposed method.
- Reviewers (rkAz, KET7) found the connection between the associative memory model (AMICL) and the proposed residual attention stream to be tenuous. Additionally, the rationale for certain design choices in the AMICL did not seem justified.
- Some issues with the plots were pointed out, and Reviewer 4pDz questioned the use of convergence speed to 100% accuracy on simple tasks as an adequate metric. Moreover, the significance of the proposed modification was not clearly demonstrated by the experiments.
- Reviewer rkAz challenged the characterization of the architecture as "biologically-inspired," arguing the link was not sufficiently established/substantiated.

In response, the authors took several steps to address most of the concerns raised by the authors:
- The authors trained and evaluated 1B sized models on a more challenging ICL task (IOI-Hard), demonstrating statistically significant though small performance improvements with their architecture.
- The writeup was revised to better explain the AMICL model and its connection to the residual value stream, as well as justifications for their design decisions, though not all reviewers were convinced.
- The authors tempered their claims regarding the biological connection and elaborated on the connections to other works pointed out by the reviewers.

While the response did not satisfy all reviewers, it did satisfy the most critical reviewer (rkAz) and me. The concern for lack of large-scale experiments is not a strong one for me, given the limitations of academic compute resources, so I discounted that. As for the evidence to support the current claims of the paper, the authors did a good job in the response to further strengthen these, and I believe adding some more synthetic experiments specifically targeting the ICL abilities (which the authors claim to be able to do) would further help. Therefore, to me the current manuscript supports its claims and acknowledges its limitations.

---

> ### Author Response · Authors · 2025-07-22
> **Minor revisions**
>
> Dear AE,
>
> Thank you for your considered and thoughtful decision.
>
> We have updated the manuscript with the suggested minor revisions. Namely, we:
> - added two paragraphs to §2.2 to further explain the normative justification and biological inspiration for the choice of the identity function in the AMICL model;
> - added an appendix and referenced this in §3.1 to provide results from our additional control experiments;
> - extended §6.3 from “Future work” to “Limitations and future work”, and described the limitations of this study in more detail;
> - added §6.2, to more clearly explain the representational benefit of the modification; and
> - checked all experimental results and reported statistical significance where appropriate.
>
> Please inform us if these changes are satisfactory. If so, we will follow-up by posting the camera-ready version (with correct formatting and author information).
>
> Kind regards,
>
> Authors

---

> > ### Author Response · Authors · 2025-07-26
> > **Camera ready version**
> >
> > Dear AE,
> >
> > We received an automated message that we should upload the camera ready version, i.e., deanonymized. Please let us know if this is incorrect, otherwise we will proceed to upload the deanonymized, camera ready version.
> >
> > Best,
> >
> > Authors

---

> > > ### Comment · Action_Editor_i3q3 · 2025-08-04
> > >
> > > Thanks for addressing the comments. I have approved the camera-ready.